# DELIBERATE REASONING FOR LLMS AS STRUCTURE-AWARE PLANNING WITH ACCURATE WORLD MODEL

## ABSTRACT

Enhancing the reasoning capabilities of large language models (LLMs) remains a key challenge, especially for tasks that require complex, multi-step decision-making. Humans excel at these tasks by leveraging deliberate planning with an internal world model to simulate the potential outcomes of various actions. Inspired by this, we propose a novel multi-step reasoning framework for LLMs, referred to as **S**tructure-a**wa**re **P**lanning with Accurate World Model (SWAP). Unlike previous approaches that rely solely on Chain-of-Thought (CoT) reasoning in natural language, SWAP incorporates structural information to guide the reasoning process via a world model and provides a soft verification mechanism over the steps. Moreover, SWAP overcomes the challenge of accurate world state predictions in complex reasoning tasks by introducing a Generator-Discriminator architecture, which enables more reliable world modeling. Specifically, the generator predicts the next state, and the discriminator ensures alignment with the logical consistency required by the problem context. SWAP also encourages the policy model to explore a broad range of potential actions to prevent premature convergence. By resolving the bottlenecks of generation diversity for both actions and states using diversity-based modeling (DBM) and improving discrimination accuracy through contrastive ranking (CR), SWAP significantly enhances the reasoning performance of LLMs. We evaluate SWAP across diverse reasoning-intensive benchmarks including math reasoning, logical reasoning, and coding tasks. Extensive experiments demonstrate that SWAP achieves substantial improvements over the baselines and consistently outperforms existing methods.

## 1 INTRODUCTION

Large language models (LLMs) (OpenAI & et al., 2024; Dubey et al., 2024) have made remarkable progress in many fields. However, their ability to perform complex reasoning remains limited (Huang & Chang, 2023). Achieving human-level problem solving is viewed as the next milestone in Artificial General Intelligence (AGI) (OpenAI, 2024). Unlike human cognition, the inference time of LLMs for reasoning tasks primarily depends on the number of input and output tokens rather than the complexity of the problem. For instance, while humans require multiple attempts, calculations, and verification to solve difficult math problems, LLMs immediately begin generating responses after reading the question. This indicates that they are not actually "thinking" but merely using intuition, *i.e.,* predicting the next token based on previous ones. In fact, there are two systems of thinking in human mind (Kahneman, 2011): System 1 operates automatically and quickly, with little effort and no sense of voluntary control; and System 2 allocates attention to the effortful mental activities that demand it. In this paper, we aim to enhance the complex reasoning capabilities of LLMs, *i.e.,* turning thinking time into better outcome, with a planning-based approach that emulates System 2.

Recently, planning and decision-making frameworks (Yao et al., 2022) have been introduced into reasoning tasks for LLMs, where the model is required not only to propose actions but also to make adjustments based on feedback from the environment. However, in many real-world scenarios, environment feedback is either unavailable or difficult to scale. Inspired by human perception (Johnson-Laird, 1983; 2010), an internal world model is introduced to enable the model to simulate actions and their effects on the world state for deliberate planning (LeCun, 2022). Some recent approaches have demonstrated success in planning and reasoning tasks with a world model (Guan et al., 2023; Hao et al., 2023), which is implemented by prompting the same LLM with in-context demonstra-

tions. However, their system performance still falls short of expectations in complex tasks, since constructing an accurate world model is inherently challenging. The predicted future state from an inaccurate world model may lead to sub-optimal or even incorrect decisions. To address this limitation, we fine-tune the model using a Generator-Discriminator architecture. Furthermore, we achieve substantial improvements on a diverse set of reasoning benchmarks by resolving the bottlenecks of generation diversity and discrimination accuracy.

On the other hand, the Chain-of-Thought (CoT) approach (Wei et al., 2022), due to its high flexibility and scalability, is widely adopted to enhance the reasoning capability of LLMs. However, it relies purely on natural language, lacking an effective verification mechanism. To address this issue, formal methods have been proposed, such as using first-order logic (Pan et al., 2023) or programs (Chen et al., 2022). Nevertheless, these formal methods are often limited in their expressiveness for a variety of tasks (Yang et al., 2024). In this paper, we propose a semi-formal approach that introduces structural information into the reasoning process, which provides a soft verification mechanism for CoTs. These structures (Dalvi et al., 2021) describe how given premises are used to generate intermediate conclusions that help validate the correctness of a particular answer. In our framework, the multi-step reasoning process involves constructing a structure, *i.e.,* the policy model proposes actions, and the world model predicts the next state and updates the structure. Specifically, new statements in the next state are introduced and linked to existing ones through entailment relations. When the reasoning is complete, the system has built an entailment graph from the given premises to the final answer, which itself serves as a justification of the reasoning process.

Specifically, our contributions mainly include:

- We introduce **structure-aware planning**, which incorporates entailment graphs into multi-step reasoning tasks. These graphs demonstrate how premises lead to intermediate conclusions and validate the correctness of the final answer, adding coherence and logical verification to the reasoning process.

- Our framework, SWAP, augments the LLM with an **accurate world model**, which is implemented using a Generator-Discriminator architecture. In addition, we resolve the bottlenecks of generation diversity and discrimination accuracy with diversity-based modelling and contrastive process supervision, respectively.

- Experiments on a diverse set of benchmarks, including math reasoning, logical reasoning and coding, show that SWAP is a general framework that achieves substantial improvements over recent popular reasoning and planning methods for LLMs.

## 2 RELATED WORK

Existing works that use advanced planning methods to enhance the multi-step problem-solving capabilities of LLMs can be categorized into three types: re-ranking (Ni et al., 2023; Wang et al., 2023b; Li et al., 2023; Lei et al., 2024), iterative correction (Madaan et al., 2023; Shinn et al., 2023; Yao et al., 2022; Chen et al., 2024a) and tree search (Chaffin et al., 2022; Gu et al., 2023; Hao et al., 2023; Yao et al., 2023; Zhou et al., 2023). Despite differences in their design, all these methods fundamentally rely on a **discriminator** to evaluate the planning steps. Recent research (Huang et al., 2023; Chen et al., 2024b) has demonstrated that the discriminator plays a more crucial role than the planning methods themselves. Consequently, using **in-context learning** to prompt the same LM as both generator and discriminator may **not sufficiently** improve the model performance on complex reasoning tasks.

To address this issue, prior research has explored various methodologies for designing the discriminator (or reward model). There are two primary types of reward models: Outcome Reward Model (ORM) and Process Reward Model (PRM). The ORM evaluates the fully generated solution by assigning a single scalar confidence score. Its training relies on outcome supervision by comparing generated answers with the ground truth. In contrast, the PRM (Lightman et al., 2023; Yuan et al., 2024; Tian et al., 2024) provides stepwise rewards throughout the reasoning process, assigning a scalar confidence score to each intermediate steps. Empirical evidence shows that, compared with outcome supervision, process supervision ensures the correctness of each step, providing more benefits to multi-step reasoning (Lightman et al., 2023). However, the training of PRM requires process supervision, which is hard to obtain, *e.g.,* collecting process annotation from humans is

inherently not scalable. Although recent research (Wang et al., 2023a; Luo et al., 2024) has increasingly explored automatic process annotations using tree search, training an effective PRM remains challenging, as from a mathematical perspective, it assigns a **numerical value** within $[0, 1]$ to each state **independently**. To overcome this problem, we propose a novel strategy for **automatic ranking annotation**, *i.e.,* given the current context and a set of candidate options, selecting the best option based on relative quality. Our ranking strategy offers significant advantages over traditional PRMs: 1) it emphasizes relative quality, making it more robust to noise; 2) it simplifies optimization and enhances generalization. Notably, our high-quality automatic ranking annotation method is non-trivial as it systemically incorporates three key factors: 1) **structural information**; 2) **correctness**; and 3) **semantical equivalence**.

Furthermore, we notice that although some reasoning processes are inherently **non-linear**, existing methods mainly follow a linear problem-solving manner. Language models are expected to implicitly infer the non-linear structure from the linear representation of the reasoning process, which proves challenging for complex reasoning tasks (Ribeiro et al., 2023). To help the model, we integrate **structural information** into the reasoning process which explicitly represents the reasoning structure within the context. These structures provide the language model with additional **guidance** and **control**, enabling extra capabilities such as symbolic learning and verification.

## 3 PRELIMINARIES

### 3.1 TASK FORMULATION

When solving complex reasoning tasks that require multiple steps, LLMs must plan intelligently, anticipating future state and guiding their reasoning towards the desired outcome. We formulate this task as a Markov Decision Process (MDP) represented by $(\mathcal{S}, \mathcal{A}, \mathcal{P}, score)$ in which:

- **State** $s_t \in \mathcal{S}$: Represents the current state, *i.e.,* all known or inferred information in the reasoning process. The initial state $s_0$ is extracted from the given context.

- **Action** $a_t \in \mathcal{A}$: Denotes a single action (produced by policy generator $\mathcal{P}_{\pi_G}$), *i.e.,* deriving new information or making inference based on current state, resulting in a state transition.

- **Transition probability** $\mathcal{P}(s_{t+1}|s_t, a_t)$: Describes the probability of transitioning to the next state $s_{t+1}$ after taking action $a_t$ in state $s_t$. We construct an enhanced world model (with generator $\mathcal{P}_{wmG}$ and discriminator $\mathcal{P}_{wmD}$) to simulate the state change.

- **Scoring function** $score(a_t|s_t)$: Quantifies the quality of an action $a_t$ given current state $s_t$. This function guides the reasoning process by prioritizing actions that are more likely to yield correct final answers. We adopt a ranking-based approach (with policy discriminator $\mathcal{P}_{\pi_D}$) instead of assigning explicit numerical scores (with PRM).

This MDP framework provides a foundation for applying planning methods to enhance the multi-step reasoning capabilities of LLMs. Each reasoning step is viewed as a decision-making process, where the model generates the next action based on current state. By updating their parameters, the models gradually learn the optimal policy for each state, improving the overall performance of reasoning. Additionally, the policy must balance exploiting known optimal actions and exploring new action spaces, guided by the scoring function to help the model make the best choices.

### 3.2 STRUCTURED REASONING AS ENTAILMENT GRAPH CONSTRUCTION

The key innovation that distinguishes our approach from related work is conceptualizing the multi-step reasoning process $(s_0, a_0, s_1, \cdots, a_{T-1}, s_T)$ as **entailment graph** (Dalvi et al., 2021) **construction** (Figure 1), which outlines how the premises in $s_0$ lead to intermediate conclusions, ultimately validating the final answer in $s_T$. Formally, let $\mathcal{G} = (\mathcal{V}, \mathcal{E})$ represent the structure, where $\mathcal{V}$ is the set of nodes, with each node $v \in \mathcal{V}$ representing a statement, *e.g.,* evidence, an assumption, or a lemma/rule; $\mathcal{E}$ is the set of directed (hyper) edges, where each (hyper) edge $e = (\mathcal{V}_{src}, \mathcal{V}_{tgt}) \in \mathcal{E}$ represents an entailment relation from a source node set $\mathcal{V}_{src} \subseteq \mathcal{V}$ (the premises) to a target node set $\mathcal{V}_{tgt} \subseteq \mathcal{V}$ (the conclusions).

Given $s_0$, the world model generator $\mathcal{P}_{wmG}$ first builds the initial graph $\mathcal{G}_0$ by extracting key statements and their relations. During the reasoning, $\mathcal{P}_{wmG}$ incrementally grows the graph by adding new

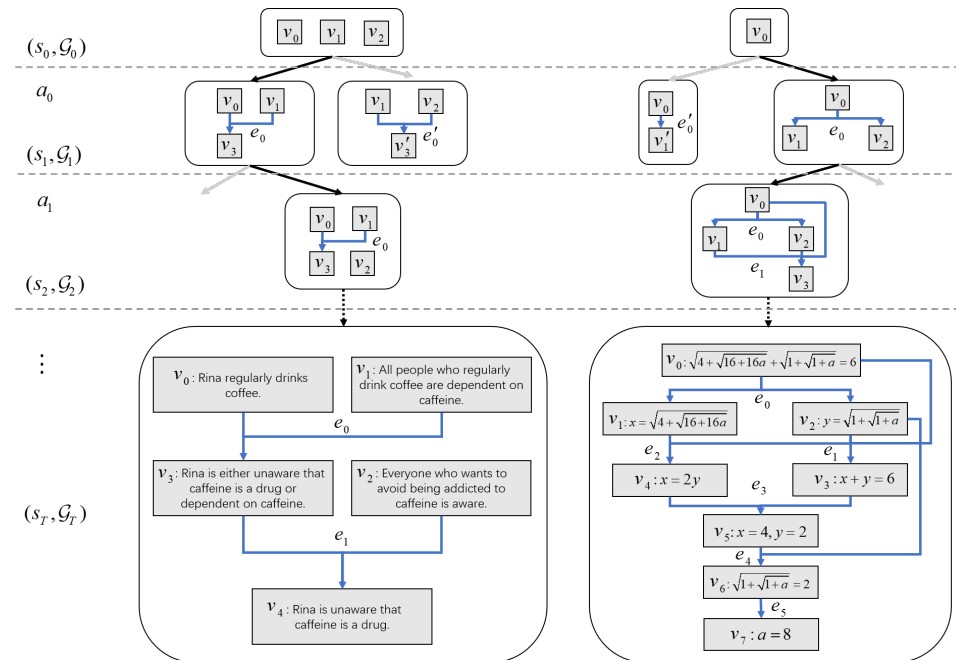

Figure 1: SWAP performs multi-step reasoning through structure-aware planning in FOLIO (left) and MATH (right). At each step, given the current state, represented as a graph, and an action, the world model predicts the next state as an updated graph.

nodes and edges, ultimately forming $\mathcal{G}_T$, which includes the final answer. The generation process is sampling-based, with the world model discriminator $\mathcal{P}_{\mathrm{wmD}}$ making decision at each step. **Structural verification** is also introduced to ensure the quality of the graph. For simplicity, let us denote the state with structural information as $(s, \mathcal{G})$. Incorporating this structure provides two main benefits: 1) the policy model can make more informed decisions using the structural information; and 2) the world model can predict more accurate next state.

# 4 STRUCTURE-AWARE PLANNING WITH ACCURATE WORLD MODEL

## 4.1 FRAMEWORK

In this section, we present the Structure-aware Planning with Accurate World Model framework (SWAP) that enables LLMs to systematically construct and utilize an entailment graph for solving a wide range of reasoning tasks. We use $\mathcal{P}_{\pi_{\mathrm{G}}}$ and $\mathcal{P}_{\pi_{\mathrm{D}}}$ to denote the policy generator & discriminator, $\mathcal{P}_{\mathrm{wmG}}$ and $\mathcal{P}_{\mathrm{wmD}}$ to denote the world model generator & discriminator, and $\mathcal{P}_{\mathrm{c}}$ to denote the controller based on pre-trained LMs. We consider $Q, A, G, H, s, \mathcal{G}, a$ as language sequences, *i.e.*, $Q = (Q[1], \cdots, Q[L])$ where each $Q[l]$ is a token, so that $\mathcal{P}(Q) = \prod_{l=1}^{L} \mathcal{P}(Q[l]|Q[1..l-1])$. We use $(s, \mathcal{G})$ to denote the state with structural information, and $c = (G, H, s, \mathcal{G})$ to denote the context of goal $G$, plan $H$ and state $(s, \mathcal{G})$.

For notational convenience, we define the generation process as gen(model, input, $N$) where $N$ is the number of generations, and the discrimination process as dis (model, input, $b$) where $b$ is the number of preserved candidates. To search potential plans and actions, we simulate the future situations of the current state using the world model. Specifically, we use sim($c, t$) to denote the simulation starting from $(s, \mathcal{G})$ up to step $t$ given the goal $G$ and plan $H$ from the context $c$.

Algorithm 1 outlines the **workflow**. Given a reasoning question $Q$, the world model generator $\mathcal{P}_{\mathrm{wmG}}(G, s_0, \mathcal{G}_0|Q)$ first generates the goal $G$ and the initial state $(s_0, \mathcal{G}_0)$. The policy generator then proposes a set of plans $H$ by sampling $N$ times from $\mathcal{P}_{\pi_{\mathrm{G}}}(H|G, s_0, \mathcal{G}_0)$. The top $b$ candidate plans are selected by the policy discriminator $\mathcal{P}_{\pi_{\mathrm{D}}}$ based on the simulation results $(s_T, \mathcal{G}_T)$ under each plan. Given the goal $G$, selected plan $H$ and current state $(s_{t-1}, \mathcal{G}_{t-1})$, multi-step reasoning at step $t$ begins with the policy generator sampling $N$ times from $\mathcal{P}_{\pi_{\mathrm{G}}}(a_{t-1}|G, H, s_{t-1}, \mathcal{G}_{t-1})$ as

**Algorithm 1** SWAP $(Q, \mathcal{P}_{\pi_{\mathrm{G}}}, \mathcal{P}_{\pi_{\mathrm{D}}}, \mathcal{P}_{\mathrm{wmG}}, \mathcal{P}_{\mathrm{wmD}}, \mathcal{P}_{\mathrm{c}}, N, T, b)$

**Require:** Reasoning question $Q$, policy generator & discriminator $\mathcal{P}_{\pi_{\mathrm{G}}}$ and $\mathcal{P}_{\pi_{\mathrm{D}}}$, world model generator & discriminator $\mathcal{P}_{\mathrm{wmG}}$ and $\mathcal{P}_{\mathrm{wmD}}$, controller $\mathcal{P}_{\mathrm{c}}$, generation number limit $N$, step limit $T$, breadth limit $b$.
  $\mathcal{D} \leftarrow \{\}$
  $G, s_0, \mathcal{G}_0 \leftarrow \mathrm{gen}(\mathcal{P}_{\mathrm{wmG}}, Q, 1)$
  $\mathcal{C} \leftarrow \{(G, H, s_0, \mathcal{G}_0) \mid H \in \mathrm{gen}(\mathcal{P}_{\pi_{\mathrm{G}}}, (G, s_0, \mathcal{G}_0), N)\}$
  $\mathcal{C} \leftarrow \mathrm{dis}(\mathcal{P}_{\pi_{\mathrm{D}}}, \{\mathrm{sim}(c, T) \mid c \in \mathcal{C}\}, b)$
  **for** $t = 1, \cdots, T$ **do**
    **if** $b = 0$ **then break**
    **end if**
    $\mathcal{C} \leftarrow \{(G, H, s, \mathcal{G}, a) \mid (G, H, s, \mathcal{G}) \in \mathcal{C}, a \in \mathrm{gen}(\mathcal{P}_{\pi_{\mathrm{G}}}, (G, H, s, \mathcal{G}), N)\}$
    $\mathcal{C} \leftarrow \mathrm{dis}(\mathcal{P}_{\pi_{\mathrm{D}}}, \{\mathrm{sim}(c, t) \mid c \in \mathcal{C}\}, b)$
    **StatePredict**$(\mathcal{C}, \mathcal{D}, \mathcal{P}_{\mathrm{wmG}}, \mathcal{P}_{\mathrm{wmD}}, \mathcal{P}_{\mathrm{c}}, N)$
  **end for**
  $A^* \leftarrow \mathrm{dis}(\mathcal{P}_{\pi_{\mathrm{D}}}, \mathcal{D}, 1)$
  **return** $A^*$

**Algorithm 2** StatePredict $(\mathcal{C}, \mathcal{D}, \mathcal{P}_{\mathrm{wmG}}, \mathcal{P}_{\mathrm{wmD}}, \mathcal{P}_{\mathrm{c}}, N)$

**Require:** Context pool $\mathcal{C}$, completed pool $\mathcal{D}$, world model generator & discriminator $\mathcal{P}_{\mathrm{wmG}}$ and $\mathcal{P}_{\mathrm{wmD}}$, controller $\mathcal{P}_{\mathrm{c}}$, generation number limit $N$.
  **parallel for** $i = 1, \cdots, b$ **do**
    $(G, H, s, \mathcal{G}, a) = \mathcal{C}_i$
    $\{(s'_j, \mathcal{G}'_j)\}_{j=1}^N \leftarrow \mathrm{gen}(\mathcal{P}_{\mathrm{wmG}}, (s, \mathcal{G}, a), N)$
    $(s', \mathcal{G}') \leftarrow \mathrm{dis}(\mathcal{P}_{\mathrm{wmD}}, \{(s, \mathcal{G}, a, s'_j, \mathcal{G}'_j)\}_{j=1}^N, 1)$
    $A \leftarrow \mathrm{gen}(\mathcal{P}_{\mathrm{c}}, (G, s', \mathcal{G}'), 1)$
    **if** $A \neq None$ **then**
      $\mathcal{D}.\mathrm{add}((G, s', \mathcal{G}', A))$ ▷ collect the state
      $\mathcal{C}.\mathrm{pop}(i)$      ▷ remove context $\mathcal{C}_i$
      $b \leftarrow b - 1$
    **else**
      $\mathcal{C}_i \leftarrow (G, H, s', \mathcal{G}')$ ▷ update context $\mathcal{C}_i$
    **end if**
  **end for**

the next action pool. The policy discriminator $\mathcal{P}_{\pi_{\mathrm{D}}}$ then evaluates and selects the top $b$ candidate contexts $(G, H, s_{t-1}, \mathcal{G}_{t-1}, a_{t-1})$ based on simulated states $(s_t, \mathcal{G}_t)$.

Then the **accurate state prediction** (Algorithm 2) is performed in parallel for each selected context $(G, H, s_{t-1}, \mathcal{G}_{t-1}, a_{t-1})$. Specifically, the world model generator predicts the next state $(s_t, \mathcal{G}_t)$ by sampling $N$ times from $\mathcal{P}_{\mathrm{wmG}}(s_t, \mathcal{G}_t \mid s_{t-1}, \mathcal{G}_{t-1}, a_{t-1})$. Then the world model discriminator $\mathcal{P}_{\mathrm{wmD}}$ selects the top 1 candidate state. Based on the selected $(s_t, \mathcal{G}_t)$, the controller determines whether to continue reasoning. If reasoning is complete, the controller $\mathcal{P}_{\mathrm{c}}(A \mid G, s_t, \mathcal{G}_t)$ generates the final answer $A$, stores $(G, s_t, \mathcal{G}_t, A)$ in the completed pool $\mathcal{D}$, and reduces $b$ by 1. Otherwise, $(G, H, s_t, \mathcal{G}_t)$ will be added to the context pool $\mathcal{C}$ for the next step. The process continues until the step limit $T$ is reached or $b$ becomes 0. Finally, the top answer $A^*$ is selected by the policy discriminator $\mathcal{P}_{\pi_{\mathrm{D}}}$ based on the completed states (with graphs) in $\mathcal{D}$.

### 4.2 SEEKING DIVERSITY IN ACTION GENERATION AND STATE PREDICTION

We identify two critical bottlenecks (generation diversity and discrimination accuracy) for the Generator-Discriminator (G-D) architecture in SWAP. Improving generation diversity is essential to allow the model to explore a broader solution space, increasing the chances of discovering the global optimal solution. Thus, we propose a **Diversity-based Modelling** (DBM) approach (Figure 2). The key idea is to encourage the generator to produce steps that differ from existing ones, thereby mitigating its inherent self-bias and promoting exploration. Compared to related work (Vijayakumar et al., 2016; Hu et al., 2023), DBM offers several advantages: 1) It builds on SFT, enabling an end-to-end learning and scalable to large datasets; 2) It leverages the extensive world knowledge embedded in pre-trained LMs.

**Diversity-based Modeling (DBM):**

Given the current state $(s_t, \mathcal{G}_t)$, we use $\mathcal{P}_{\pi_{\mathrm{G}}}^{\mathrm{ori}}(a_t \mid G, H, s_t, \mathcal{G}_t)$ to denote the original distribution learned by supervised fine-tuning on the positive trajectories (that lead to correct final answers) during training. For $n$-th generation, we aim to introduce diversity by considering an additional distribution $\mathcal{P}_{\pi_{\mathrm{G}}}^{\mathrm{sem}}(a_t^n \mid a_t^{1..n-1})$, which represents steps that are semantically similar to those generated previously $a_t^{1..n-1}$. Specifically, the probability of $l$-th token $a_{t,l}^n$ in the $n$-th generation $a_t^n$ is

$$\mathcal{P}_{\pi_{\mathrm{G}}}^{\mathrm{sem}}(a_{t,l}^n \mid a_t^{1..n-1}, a_{t,1..l-1}^n) = \frac{1}{n-1} \sum_{j=1}^{n-1} \mathcal{P}_{\pi_{\mathrm{G}}}^{\mathrm{sem}}(a_{t,l}^n \mid a_t^j, a_{t,1..l-1}^n), \tag{1}$$

where $a_t^j$ denotes the $j$-th generation, and for notational simplicity, we **move the token index to a subscript**, so that $a_{t,1..l-1}^n$ denotes the preceding tokens of the $l$-th token $a_{t,l}^n$. We obtain $\mathcal{P}_{\pi_G}^{\text{sem}}(a_l'|a, a_{1..l-1}')$ by using supervised fine-tuning on the training data generated by GPT-4o, where $a$ and $a'$ are pairs of actions that are semantically equivalent.

To encourage diversity, the generator adjusts the original distribution $\mathcal{P}_{\pi_G}^{\text{ori}}$ by reducing the probability mass assigned to steps that are semantically similar to previous generations, *i.e.,*

$$\mathcal{P}_{\pi_G}(a_{t,l}^n|G, H, s_t, \mathcal{G}_t, a_t^{1..n-1}, a_{t,1..l-1}^n) = \text{Norm}\bigg(\mathcal{P}_{\pi_G}^{\text{ori}}(a_{t,l}^n|G, H, s_t, \mathcal{G}_t, a_{t,1..l-1}^n)$$

$$- \gamma_l \mathcal{P}_{\pi_G}^{\text{sem}}(a_{t,l}^n|a_t^{1..n-1}, a_{t,1..l-1}^n)\bigg) \qquad (2)$$

where the decay factor, $\gamma_l = \gamma_0 \cdot \alpha^l$ with $\alpha \leq 1$, is introduced to emphasize diversity in early stages of generation while gradually reducing this effect. This ensures that the deduplication effect is stronger initially to explore different paths but weakens over time to avoid drifting too far from plausible solutions, thereby maintaining accuracy. Note that this discussion primarily focuses on action generation, while the process of plan generation using the policy generator, represented as $\mathcal{P}_{\pi_G}(H^n|G, s_0, \mathcal{G}_0, H^{1..n-1})$, follows a similar approach.

The normalization function

$$\text{Norm}(\mathcal{P}) = \frac{\max(\mathcal{P}, 0)}{\mathbf{1}^\top \max(\mathcal{P}, 0)} \qquad (3)$$

is applied to discard negative-valued tokens (that resemble previous generations or deviate from the intended progression of reasoning) and maintain a diverse and relevant generation. Other alternatives, such as Softmax, can distort the probability of irrelevant tokens by redistributing values across all tokens.

**State Prediction Enhancement with Diversity:**

By encouraging the generator to produce diverse predictions, we increase the likelihood of overcoming self-biases and discovering a more accurate future state. We then select the top 1 prediction from the diverse options generated. To achieve this, we apply a similar strategy to enhance diversity for state prediction $\mathcal{P}_{\text{wmG}}(s_t^n|s_{t-1}, \mathcal{G}_{t-1}, a_{t-1}, s_t^{1..n-1})$, that is,

$$\mathcal{P}_{\text{wmG}}(s_{t,l}^n|s_{t-1}, \mathcal{G}_{t-1}, a_{t-1}, s_t^{1..n-1}, s_{t,1..l-1}^n) = \text{Norm}\bigg(\mathcal{P}_{\text{wmG}}^{\text{ori}}(s_{t,l}^n|s_{t-1}, \mathcal{G}_{t-1}, a_{t-1}, s_{t,1..l-1}^n)$$

$$- \gamma_l \mathcal{P}_{\text{wmG}}^{\text{sem}}(s_{t,l}^n|s_t^{1..n-1}, s_{t,1..l-1}^n)\bigg) \qquad (4)$$

where $s_t^j$ denotes the $j$-th generation, and $s_{t,1..l-1}^n$ is the preceding tokens of the $l$-th token $s_{t,l}^n$. Once the state $s_t^n$ is generated, the corresponding graph $\mathcal{G}_t^n$ is extracted from this state, allowing the model to maintain a consistent representation of entailment relationships as the reasoning progresses.

In addition to diversity-based modeling, we leverage a **dynamic context strategy** to further diversify the generation. This strategy involves randomly reframing the current state to create an alternative

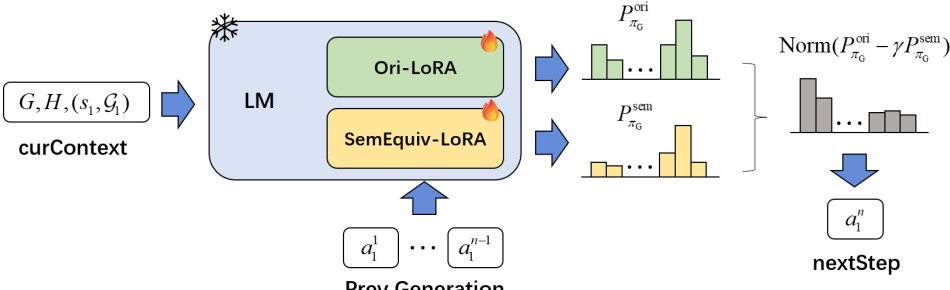

Figure 2: Overview of the proposed Diversity-Based Modeling (DBM) method. The current context is processed by the language model, which is fine-tuned using Ori-LoRA and SemEquiv-LoRA. Previous generations are used to compute the semantic equivalence distribution, which is employed to adjust the original distribution to avoid repetition.

context for each step. For example, given the original state $(s, \mathcal{G})$, we generate an alternative state $(s', \mathcal{G}')$, where $s'$ is sampled from the semantic equivalence distributions $\mathcal{P}^{\text{sem}}_{\text{wmG}}(s'|s)$. The corresponding graph $\mathcal{G}'$ is then regenerated from $s'$. Our experiments show that this strategy significantly contributes to generating diverse outputs, enhancing the model's robustness and performance on reasoning tasks.

### 4.3 IMPROVING DISCRIMINATION ACCURACY IN REASONING

As highlighted in recent works (Huang et al., 2023; Chen et al., 2024b), discrimination accuracy is a critical aspect of various planning methods. However, training an effective PRM remains challenging, as mathematically, it assigns a numerical value to each state independently. To address this issue, our discriminator employs **Contrastive Ranking** (CR) to evaluate multiple candidate options simultaneously. By focusing on relative comparisons, the model can effectively identify discrepancies between options, particularly erroneous parts, thereby simplifying the task.

**Contrastive Ranking (CR) for Enhanced Evaluation:**

To illustrate (Figure 3), given a positive trajectory $[(s_0, \mathcal{G}_0), a_0, \cdots, (s_T, \mathcal{G}_T)]$ that leads to the correct final answer, we randomly select an intermediate step $t$, and finalize $K$ subsequent reasoning processes: $\{[a_t^j, \cdots, (s_{T_j}^j, \mathcal{G}_{T_j}^j)]\}_{j=1}^K$, where $T_j$ represents the length of the $j$-th trajectory. Among these $K$ trajectories, we identify the first erroneous steps in negative trajectories (which lead to incorrect final answers) by determining which steps are semantically different from the positive trajectory and then performing **structural verification** and $N_{\text{veri}}$ completions for **outcome verification**, *i.e.,* if none of the completions result in the correct final answer, we confirm these steps as erroneous.

Given the contrastive process annotations, we define the inputs and outputs of the discriminator while incorporating **meta knowledge** $\mathcal{K}_{\text{meta}}$ to enhance model performance. Specifically,

$$E, a_t^{\text{best}} \sim \mathcal{P}_{\pi_{\text{D}}}\left(E, a_t^{\text{best}} \mid \mathcal{K}_{\text{meta}}, G, H, (s_t, \mathcal{G}_t), \{a_t^j, (s_{t+1}^j, \mathcal{G}_{t+1}^j)\}_{j=1}^K\right) \tag{5}$$

$$E, s_{t+1}^{\text{best}}, \mathcal{G}_{t+1}^{\text{best}} \sim \mathcal{P}_{\text{wmD}}\left(E, s_{t+1}^{\text{best}}, \mathcal{G}_{t+1}^{\text{best}} \mid \mathcal{K}_{\text{meta}}, (s_t, \mathcal{G}_t), a_t, \{(s_{t+1}^j, \mathcal{G}_{t+1}^j)\}_{j=1}^K\right) \tag{6}$$

where $(s_{t+1}^j, \mathcal{G}_{t+1}^j)$ is used for the selection of action $a_t$. We avoid using longer future trajectories to prevent introducing new errors, which could interfere with action selection. For plan selection $\mathcal{P}_{\pi_{\text{D}}}(E, H^{\text{best}} \mid \mathcal{K}_{\text{meta}}, G, (s_0, \mathcal{G}_0), \{H^j, (s_{T_j}^j, \mathcal{G}_{T_j}^j)\}_{j=1}^K)$, we use the simulated completed states $(s_{T_j}^j, \mathcal{G}_{T_j}^j)$ for the selection of $H$. The discriminator generates an explanation $E$, highlighting differences between the $K$ future states before making a decision. We fine-tune the discriminator using these explanations through bootstrapping from GPT-4o. We use the superscript 'best' to denote the final selected option, *i.e.,* $H^{\text{best}}$, $a_t^{\text{best}}$ and $(s_{t+1}^{\text{best}}, \mathcal{G}_{t+1}^{\text{best}})$, and the construction of meta knowledge $\mathcal{K}_{\text{meta}}$ based on training data is provided in Appendix A.

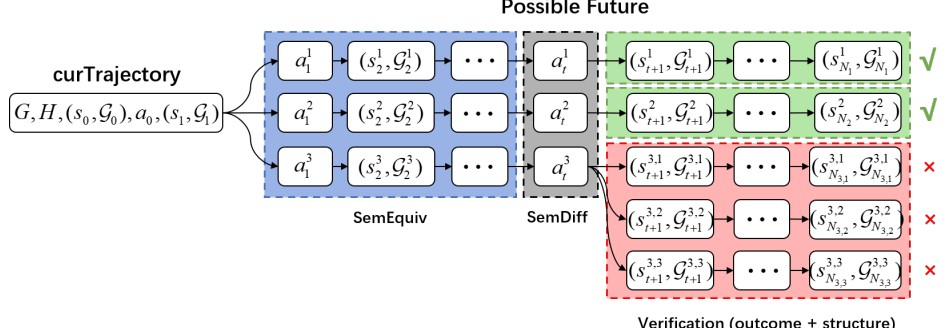

Figure 3: Overview of our automatic ranking annotation. Starting from a selected step in the positive trajectory, multiple future actions and states are generated to create candidate trajectories. Negative trajectories, which lead to incorrect final answers, are analyzed to identify the first steps that are semantically different from those in the positive trajectory. Structural verification and tree search for outcome are then employed to identify these potential erroneous steps.

Table 1: Overall performance comparison across different benchmark datasets. The best performance for each task using the same base model is in bold. Note: We use the filtered PRM800K dataset (Sun et al., 2024) to evaluate performance on the full MATH test set.

| Model | Math Reasoning | | Logical Reasoning | | Coding | |
|---|---|---|---|---|---|---|
| | GSM8K | MATH | FOLIO | ReClor | HumanEval | MBPP |
| LLaMA3-8B-Instruct | | | | | | |
| Zero-shot CoT | $70.0 \pm 2.0$ | $27.6 \pm 0.6$ | $62.1 \pm 1.8$ | $57.8 \pm 1.4$ | $53.3 \pm 0.6$ | $51.8 \pm 0.2$ |
| Few-shot CoT (4-shot) | $72.4 \pm 1.8$ | $23.6 \pm 0.6$ | $57.2 \pm 1.4$ | $52.1 \pm 1.1$ | $56.8 \pm 0.2$ | $53.6 \pm 0.2$ |
| SFT-CoT | $71.3 \pm 1.8$ | $25.4 \pm 0.4$ | $66.0 \pm 0.8$ | $62.2 \pm 0.8$ | $51.6 \pm 0.4$ | $51.0 \pm 0.3$ |
| Self-consistency | $74.1 \pm 1.2$ | $26.0 \pm 0.4$ | $66.2 \pm 0.5$ | $60.1 \pm 0.6$ | - | - |
| ToT | $75.2 \pm 1.1$ | $28.8 \pm 0.4$ | $67.1 \pm 0.8$ | $60.6 \pm 0.8$ | - | - |
| RAP | $76.0 \pm 1.0$ | $28.4 \pm 0.3$ | $67.5 \pm 0.6$ | $61.3 \pm 0.6$ | - | - |
| PRM (PRM800K$^*$) | $74.6 \pm 0.8$ | $28.8 \pm 0.2$ | - | - | - | - |
| PRM (Math-Shepherd) | $76.2 \pm 0.8$ | $28.6 \pm 0.3$ | - | - | - | - |
| SWAP (w/o discriminator) | $78.1 \pm 1.0$ | $37.3 \pm 0.4$ | $69.2 \pm 0.8$ | $69.1 \pm 0.8$ | $53.1 \pm 0.8$ | $53.4 \pm 0.6$ |
| SWAP | $\mathbf{82.7} \pm 0.6$ | $\mathbf{42.3} \pm 0.3$ | $\mathbf{73.2} \pm 0.5$ | $\mathbf{74.1} \pm 0.4$ | $\mathbf{57.8} \pm 0.6$ | $\mathbf{58.6} \pm 0.4$ |
| Mistral-7B-Instruct | | | | | | |
| Zero-shot CoT | $23.4 \pm 1.8$ | $12.0 \pm 0.4$ | $46.8 \pm 1.5$ | $38.8 \pm 1.0$ | $42.5 \pm 0.5$ | $38.8 \pm 0.4$ |
| Few-shot CoT (4-shot) | $47.3 \pm 1.6$ | $12.7 \pm 0.5$ | $48.6 \pm 1.6$ | $36.2 \pm 0.8$ | $43.6 \pm 0.4$ | $44.8 \pm 0.6$ |
| SFT-CoT | $48.0 \pm 1.0$ | $12.6 \pm 0.3$ | $52.0 \pm 1.0$ | $40.2 \pm 0.6$ | $43.8 \pm 0.4$ | $46.0 \pm 0.4$ |
| Self-consistency | $52.1 \pm 0.8$ | $11.2 \pm 0.2$ | $51.2 \pm 0.6$ | $42.4 \pm 0.4$ | - | - |
| ToT | $49.6 \pm 1.2$ | $12.3 \pm 0.3$ | $50.2 \pm 1.2$ | $40.8 \pm 0.8$ | - | - |
| RAP | $56.1 \pm 1.0$ | $13.0 \pm 0.2$ | $52.1 \pm 0.8$ | $41.6 \pm 0.6$ | - | - |
| PRM (PRM800K$^*$) | $54.2 \pm 0.8$ | $14.2 \pm 0.2$ | - | - | - | - |
| PRM (Math-Shepherd) | $55.4 \pm 0.6$ | $13.6 \pm 0.2$ | - | - | - | - |
| SWAP (w/o discriminator) | $54.0 \pm 0.8$ | $15.4 \pm 0.3$ | $54.0 \pm 0.6$ | $45.2 \pm 0.6$ | $45.0 \pm 0.8$ | $47.0 \pm 0.4$ |
| SWAP | $\mathbf{60.4} \pm 0.6$ | $\mathbf{18.7} \pm 0.2$ | $\mathbf{58.0} \pm 0.3$ | $\mathbf{49.1} \pm 0.4$ | $\mathbf{48.4} \pm 0.6$ | $\mathbf{51.1} \pm 0.3$ |

During inference, given the discriminator, we apply a **voting** strategy to decide the top $b$ candidates as mentioned in Algorithm 1, 2. To further enhance robustness, we reframe the candidate options and reorder them to have **multiple comparisons** within the same group. In addition, we further enhance the discrimination accuracy with **structural verification** on the graphs $\{\mathcal{G}^j\}_{j=1}^K$. Details of these strategies are given in Appendix A.

## 5 EXPERIMENTS

### 5.1 EXPERIMENTAL SETUP

We conduct experiments on various types of reasoning tasks. Dataset statistics and examples are provided in Appendix B. For each dataset, we use different types of models (GPT-4o (OpenAI & et al., 2024), DeepSeek-V2 (DeepSeek-AI & et al., 2024), LLaMA3 (Dubey et al., 2024)) to generate multiple trajectories for the training and validation sets. We label the trajectories as positive or negative based on their final answers. To improve the model stability, we augment training questions using GPT-4o. Given the positive and negative trajectories of the same question, we automatically generate contrastive process annotations (Figure 3) using DeepSeek-V2. Additionally, to address the class imbalance in contrastive ranking data, we apply pre-processing and post-processing techniques (see Appendix D for details). With the complete training data, SWAP is fine-tuned from LLaMA3-8B-Instruct using LoRA (Hu et al., 2021). The parameter settings are as follows: For DBM, $\gamma_0 = 0.7$ and $\alpha = 0.95$. For CR, $N_{\text{veri}} = 3$; we choose $K = \{2, 3\}$ for discriminator training, and during inference, multiple options are divided into groups of size 2 or 3; for meta knowledge, we use $\mathcal{M} = 5$. To ensure the effectiveness of training, we also employ specialized strategies such as curriculum learning and self-improving training (details in Appendix D). During evaluation, we compare our SWAP against popular strategies, CoT, Self-consistency (SC) (Wang et al., 2023b), ToT (Yao et al., 2023), and RAP (Hao et al., 2023)) as well as SFT on CoTs and verification with PRMs (Lightman et al., 2023; Wang et al., 2023a), using different base models (LLaMA3-8B-Instruct and Mistral-7B-Instruct (Jiang et al., 2023)). The number of candidate solutions for self-consistency and PRMs is set to 8. For SWAP, ToT, and RAP (utilizing MCTS), the generation number and step limits are set to 5 and 10, respectively. The number of rollouts (breadth limit) is set as 8. More details about data generation, model training and evaluation are provided in Appendix C, D.

## 5.2 MAIN RESULTS

Overall performance is shown in Table 1, with fine-grained results and examples provided in Appendix E and Appendix G, respectively. We summarize the key findings as follows:

**SWAP consistently achieves the best or comparable performance among different methods.** One-pass CoT and verification methods, such as self-consistency and PRMs, do not involve searching through intermediate steps during the reasoning process. In contrast, our framework empowers the model to reason more like humans' conscious planning, which significantly improves performance on multi-step reasoning tasks. This planning ability becomes especially crucial in more challenging tasks, where deliberate reasoning is necessary to avoid intermediate errors. For instance, our framework with LLaMA3-8B-Instruct achieves a 14.7% better accuracy compared to CoT (53% relative improvement) on the more difficult MATH dataset, and a 10.3% improvement on GSM8k.

**Structure-aware planning and an accurate world model further enhance the effectiveness of planning in LLMs.** Methods such as ToT and RAP, which also incorporate planning or search-based strategies, do not match our approach in performance. They lack the deeper structural understanding and precise state modeling that our framework provides. SWAP explicitly introduces the structure that describes the relationship between key statements, which facilitates both action generation and state prediction. In addition, Diversity-based Modeling (DBM) enables the generator to explore a broader solution space, increasing the likelihood of finding optimal steps. Contrastive Ranking (CR), on the other hand, significantly improves the accuracy of the discriminator by focusing on relative comparisons between candidate solutions. This combination of enhanced exploration and more precise discrimination is key to the substantial performance improvements observed in our experiments, especially on challenging datasets like MATH.

## 5.3 ANALYSIS

We investigate the effect of search tree width and depth on overall accuracy, providing insights for both parameter selection and dynamic evaluation across various tasks.

**The benefit of increasing search tree width, *i.e.*, the number of search attempts per step, becomes marginal after a certain point.** For planning-based approaches, the width of the search tree directly influences the thoroughness of exploring the solution space at each step. We analyze the effect of search width on accuracy in SWAP (Figure 4). As shown, there is a consistent upward trend across all datasets. However, the benefits diminish beyond a certain point, *e.g.*, after 5-7 search attempts in FOLIO and GSM8K, since most of the promising options have already been explored. We found that using a search width of 5 offers the best trade-off between computational cost and performance. We also observed some variability between datasets. GSM8K and MATH show a sharper initial increase in accuracy with fewer search attempts, while FOLIO and HumanEval exhibit a more gradual improvement. This discrepancy likely arises from the variations in task complexity and dataset size.

**Model performance improves gradually with increasing search tree depth, *i.e.*, the number of searched steps in the trajectory.** Another important factor is the search tree depth, which refers to the number of searched steps. We analyze how accuracy changes with search depth in SWAP (Figure 5). For each value of $N_{\text{search}}$, we search and optimize the first $N_{\text{search}}$ steps and allow the model to

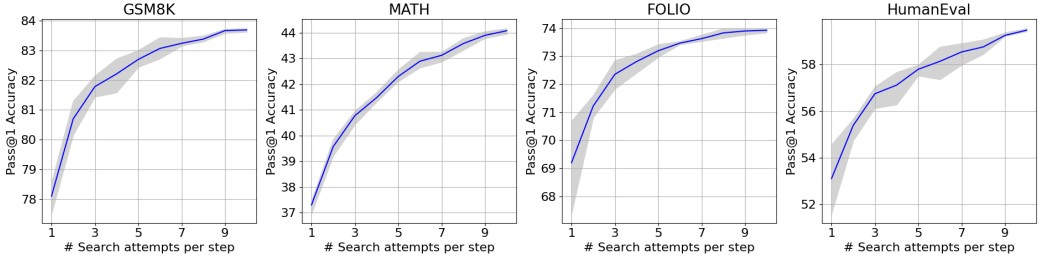

Figure 4: Effect of increasing search tree width on overall accuracy for different benchmark datasets in SWAP. The accuracy generally improves as search tree width increases, but the benefits become marginal beyond a certain number of search attempts.

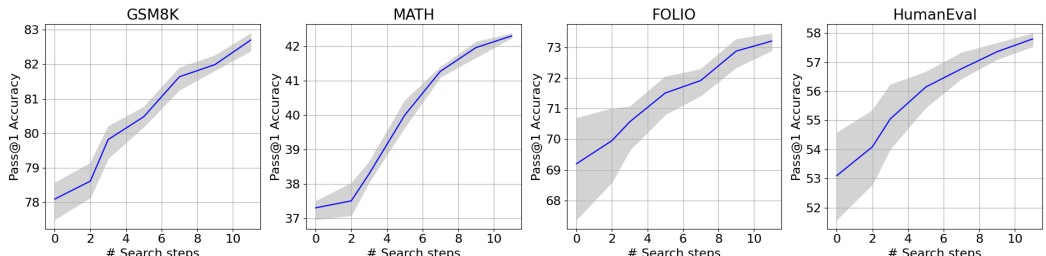

Figure 5: Effect of increasing search tree depth on overall accuracy for different benchmark datasets in SWAP. More searched steps lead to improved accuracy and reduced variance.

Table 2: Ablation studies. The complete framework achieves the highest performance across all tasks, demonstrating that each component contributes positively to overall accuracy.

| Method | Math Reasoning | | Logical Reasoning | | Coding | |
|---|---|---|---|---|---|---|
| | GSM8K | MATH | FOLIO | ReClor | HumanEval | MBPP |
| SWAP (Ours) | **82.7** | **42.3** | **73.2** | **74.1** | **57.8** | **58.6** |
| w/o structure info | 81.2 | 40.4 | 72.5 | 71.8 | 56.1 | 57.5 |
| w/o DBM | 79.0 | 38.5 | 70.0 | 71.2 | 55.0 | 55.9 |
| w/o meta knowledge | 81.3 | 40.8 | 72.4 | 73.0 | 56.2 | 57.1 |
| w/ PRM instead of CR | 81.0 | 39.1 | 71.6 | 72.0 | 55.9 | 56.8 |
| w/o discriminator | 78.1 | 37.3 | 69.2 | 69.1 | 53.1 | 53.4 |

complete the remaining trajectory directly. As seen, accuracy steadily increases with the number of searched steps across all datasets, indicating that our planning brings benefits. Notably, the benefits of planning depend on the difficulty of each stage, as more challenging steps yield greater accuracy improvements after searching. Toward the end of the trajectory, the accuracy curve begins to flatten, and its variance is reduced as the trajectory converges to the optimal one.

## 5.4 ABLATION STUDY

We analyze the impact of the key components proposed in this paper (Table 2). The complete framework achieves the highest performance across all tasks, demonstrating that each component contributes positively to overall accuracy. Notably, the discriminator has the most significant impact by effectively selecting optimal actions and state predictions. The incorporation of structural information is also crucial, particularly for complex reasoning tasks like math and logical reasoning. DBM enhances generation diversity by promoting the exploration of diverse solution paths, while CR outperforms PRM in multi-step reasoning, as selecting the optimal solution by comparing different options is more reliable than scoring each option independently. Finally, incorporating meta knowledge further improves discrimination accuracy. These improvements are consistently observed across different task types.

## 6 CONCLUSION

In this paper, we introduce SWAP, a novel framework for enhancing the multi-step reasoning capabilities of LLMs through structure-aware planning with an accurate world model. Our approach consistently outperforms existing methods in extensive experiments, demonstrating significant improvements on reasoning-heavy benchmarks, including math, logical reasoning, and coding tasks. In this work, we primarily adopt a re-ranking strategy, as it provides a good balance between computational cost and model performance. For future research, exploring reinforcement learning (RL) methods to enable dynamic interaction with the world model could further optimize LLMs for long-term rewards. Additionally, teaching the model to recognize and correct its own mistakes represents another promising direction, potentially leading to even more robust reasoning capabilities.

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

## A  ADDITIONAL METHODOLOGICAL DETAILS

The **meta knowledge** $\mathcal{K}_{\text{meta}}$, which helps verify answers and identify errors, is derived from training questions. Formally, $\mathcal{K}_{\text{meta}} = \text{concat}_m(\mathcal{K}_m)$, where $\mathcal{K}_m$ represents stored knowledge from the $m$-th training sample, and we select the top $\mathcal{M}$ samples based on the cosine similarity between the training query embedding $q_m$ and the test query embedding $q$.

We fine-tune our discriminator using the contrastive process annotations, which helps it accurately identify subtle differences between trajectories and improve its ability to distinguish between valid and invalid reasoning steps. To further enhance robustness during inference, we randomly **group the candidate options** and reorder them, then apply a **voting** strategy to determine the final ranking to decide the top $b$ candidates as mentioned in Algorithm 1, 2. This approach ensures that the model is not biased by specific sequences and provides a more reliable assessment of the best candidate.

In addition, we introduce **structural verification** for generated entailment graphs $\mathcal{G}$ to further enhance discrimination. Key steps involves: 1) Syntax Verification: Validates the format of nodes and edges. 2) Node Dependency Analysis: Examines the dependencies between nodes (assumptions, lemmas, facts, or inferences derived from prior nodes). 3) Cycle Detection: Ensures acyclic structures to maintain logical consistency. 4) Redundancy Check: Detects redundant or disconnected nodes. All of them are implemented according to standard graph algorithms.

## B  DATASET OVERVIEW

In this section, we present statistics and examples for all benchmark datasets used in our study. We consider GSM8K (Cobbe et al., 2021), MATH (Hendrycks et al., 2021) for math reasoning, FOLIO (Han et al., 2022), ReClor (Yu et al., 2020) for logical reasoning, and HumanEval (Chen et al., 2021), MBPP (Austin et al., 2021) for coding. For GSM8K, there are 7,473 training samples and 1,319 test samples. For MATH, there are 7,500 training samples and 5,000 test samples. For FOLIO, the training and validation sets consist of 1,001 and 203 samples, respectively. For ReClor, we use 4,638 training samples, 500 validation samples (used as test set, as the original test set answers are not publicly available), and 1,000 test samples. HumanEval contains 164 test samples, and since it lacks a training set, we use the entire MBPP dataset (after format transformation) for training. MBPP consists of 374 training samples, 90 validation samples, and 500 test samples.

---

**Example - GSM8K**

**Problem:** Weng earns $12 an hour for babysitting. Yesterday, she just did 50 minutes of babysitting. How much did she earn?

**Solution:**
Weng earns 12/60 = \$<<12/60=0.2>>0.2 per minute.
Working 50 minutes, she earned 0.2 x 50 = \$<<0.2*50=10>>10.
#### 10

**Problem:** Janet hires six employees. Four of them are warehouse workers who make $15/hour, and the other two are managers who make $20/hour. Janet has to pay 10% of her workers' salaries in FICA taxes. If everyone works 25 days a month and 8 hours a day, how much does Janet owe total for their wages and taxes for one month?

**Solution:**
First figure out how many hours each worker works per month by multiplying the number of days they work by the number of hours a day they work: 25 days $*$ 8 hours/day = <<25*8=200>>200 hours
Then calculate how much one warehouse worker makes per month by multiplying their hourly rate by the number of hours they work: 200 hours $*$ \$15/hour = \$<<200*15=3000>>3000
Then multiply that number by 4 to find out how much all the warehouse workers make: \$3000/worker $*$ 4 workers = \$<<3000*4=12000>>12,000
Now multiply the hours each manager works (also 200) by their hourly wage to find out how much one manager makes per month: 200 hours $*$ \$20/hour = \$<<200*20=4000>>4,000
Now multiply one manager's wages by the number of managers (2) to find their total wage amount: \$4,000/manager $*$ 2 managers = \$<<4000*2=8000>>8,000
Now add the wages for the managers and the workers to find the total cost of the wages: \$8,000 + \$12,000 = \$<<8000+12000=20000>>20,000

---

Now multiply the total wage bill by 10% to find how much the FICA taxes are: $20,000 * .1 = $
<<20000*.1=2000>>2,000
Now add the total wage bill to the total tax amount to find the grand total: $2,000 + $20,000 = $
<<2000+20000=22000>>22,000
#### 22000

## Example - MATH

**Problem:** Let $f(x) = \begin{cases} ax + 3, & \text{if } x > 2, \\ x - 5 & \text{if } -2 \leq x \leq 2, \\ 2x - b & \text{if } x < -2. \end{cases}$

Find $a + b$ if the piecewise function is continuous (which means that its graph can be drawn without lifting your pencil from the paper).

**Solution:**
For the piecewise function to be continuous, the cases must "meet" at $2$ and $-2$. For example, $ax + 3$ and $x - 5$ must be equal when $x = 2$. This implies $a(2) + 3 = 2 - 5$, which we solve to get $2a = -6 \Rightarrow a = -3$. Similarly, $x - 5$ and $2x - b$ must be equal when $x = -2$. Substituting, we get $-2 - 5 = 2(-2) - b$, which implies $b = 3$. So $a + b = -3 + 3 = \boxed{0}$.

**Problem:** Square ABCD has its center at $(8, -8)$ and has an area of 4 square units. The top side of the square is horizontal. The square is then dilated with the dilation center at (0,0) and a scale factor of 2. What are the coordinates of the vertex of the image of square ABCD that is farthest from the origin? Give your answer as an ordered pair.

**Solution:**
With the center of dilation at the origin and a scale factor of 2, all the coordinates of square $ABCD$ are twice the coordinates of its preimage. The preimage has an area of 4 square units, so its side length is 2 units. Since the center of the preimage is at $(8, -8)$, the four vertices of the preimage are at $(7, -9), (7, -7), (9, -7)$ and $(9, -9)$. The point $(9, -9)$ is the farthest from the origin on the preimage, so the point farthest from the origin on the image of square $ABCD$ is $\boxed{(18, -18)}$.

## Example - FOLIO

**Problem:**
Premises:
All customers in James' family who subscribe to AMC A−List are eligible to watch three movies every week without any additional fees.
Some of the customers in James' family go to the cinema every week.
Customers in James' family subscribe to AMC A−List or HBO service.
Customers in James' family who prefer TV series will not watch TV series in cinemas.
All customers in James' family who subscribe to HBO services prefer TV series to movies.
Lily is in James' family; she watches TV series in cinemas.

Conclusion:
Lily goes to cinemas every week or watches 3 movies every week without any additional fees.

**Solution:** True

**Problem:**
Premises:
If a legislator is found guilty of stealing government funds, they will be suspended from office.
Tiffany T. Alston was a legislator in Maryland's House of Delegates from 2011 to 2013.
Tiffany T. Alston was found guilty of stealing government funds in 2012.

Conclusion:
Tiffany T. Alston went to prison for stealing government funds.

**Solution:** Uncertain

## Example - ReClor

**Problem:**
Paula will visit the dentist tomorrow morning only if Bill goes golfing in the morning. Bill will not go golfing unless Damien agrees to go golfing too. However, Damien has decided not to go golfing. Ttherefore, Paula will not be visiting the dentist tomorrow morning.

0. If Marge goes to the bank today, Lauren will not cash her check tomorrow. Marge will not wash her car unless it is sunny. However, it is sunny, so Marge will wash her car and go shopping with Lauren.
1. Kevin will wash his car tomorrow only if Brittany has to go visit her grandmother. Unless Aunt Susan has to run errands, Brittany will not have to go visit her grandmother. Since Aunt Susan does not have to run errands, Kevin will not wash his car tomorrow.
2. Renee will do her homework tonight if there is nothing good on television and if her neighbors do not have a party. Although, there is something good on television; her neighbors are also having a party. Ttherefore, Renee will attend the party.
3. Maddie will plan a picnic only if one of her friends, Lisa or Kenny, will come. Kenny will not come to the picnic, but Lisa will. Ttherefore, Maddie will plan a picnic.

The pattern of reasoning displayed above most closely parallels which of the following?

**Solution:** 1

**Problem:**
Environmentalist: An increased number of oil spills and the consequent damage to the environment indicate the need for stricter safety standards for the oil industry. Since the industry refuses to take action, it is the national government that must regulate industry safety standards. In particular, the government has to at least require oil companies to put double hulls on their tankers and to assume financial responsibility for accidents. Industry representative: The industry alone should be responsible for devising safety standards because of its expertise in handling oil and its understanding of the cost entailed. Implementing the double−hull proposal is not currently feasible because it creates new safety issues. Furthermore, the cost would be burdensome to the industry and consumers.

0. Government safety regulations are developed in a process of negotiation with industry leaders and independent experts.
1. Environmental concerns outweigh all financial considerations when developing safety standards.
2. The requirement of two hulls on oil tankers, although initially costly, will save money over time by reducing cleanup costs.
3. The only effective sources of increased stringency in safety standards for oil tankers are action by the industry itself or national government regulation.

Which one of the following is an assumption on which the argument of the environmentalist depends?

**Solution:** 3

## Example - HumanEval

**Problem:**
```
from typing import List

def separate_paren_groups(paren_string: str) -> List[str]:
    ''' Input to this function is a string containing multiple groups of nested parentheses. Your goal is to
    separate those group into separate strings and return the list of those.
    Separate groups are balanced (each open brace is properly closed) and not nested within each other
    Ignore any spaces in the input string.
    >>> separate_paren_groups('( ) (( )) (( )( ))')
    ['()', '(())', '(()())']
    '''
```

**Solution:**
```
    result = []
    current_string = []
    current_depth = 0

    for c in paren_string:
```

```
        if c == '(':
            current_depth += 1
            current_string.append(c)
        elif c == ')':
            current_depth -= 1
            current_string.append(c)

            if current_depth == 0:
                result.append(''.join(current_string))
                current_string.clear()

    return result
```

**Problem:**
```
def count_up_to(n):
    ''' Implement a function that takes an non−negative integer and returns an array of the first n
    integers that are prime numbers and less than n.
    for example:
    count_up_to(5) => [2,3]
    count_up_to(11) => [2,3,5,7]
    count_up_to(0) => []
    count_up_to(20) => [2,3,5,7,11,13,17,19]
    count_up_to(1) => []
    count_up_to(18) => [2,3,5,7,11,13,17]
    '''
```

**Solution:**
```
    primes = []
    for i in range(2, n):
        is_prime = True
        for j in range(2, i):
            if i % j == 0:
                is_prime = False
                break
        if is_prime:
            primes.append(i)
    return primes
```

---

## Example - MBPP

**Problem:**
Write a function to find the minimum cost path to reach (m, n) from (0, 0) for the given cost matrix cost[][]
and a position (m, n) in cost[][].

```
assert min_cost([[1, 2, 3], [4, 8, 2], [1, 5, 3]], 2, 2) == 8
assert min_cost([[2, 3, 4], [5, 9, 3], [2, 6, 4]], 2, 2) == 12
assert min_cost([[3, 4, 5], [6, 10, 4], [3, 7, 5]], 2, 2) == 16
```

**Solution:**
```
R = 3
C = 3
def min_cost(cost, m, n):
    tc = [[0 for x in range(C)] for x in range(R)]
    tc[0][0] = cost[0][0]
    for i in range(1, m+1):
        tc[i][0] = tc[i−1][0] + cost[i][0]
    for j in range(1, n+1):
        tc[0][j] = tc[0][j−1] + cost[0][j]
    for i in range(1, m+1):
        for j in range(1, n+1):
            tc[i][j] = min(tc[i−1][j−1], tc[i−1][j], tc[i][j−1]) + cost[i][j]
    return tc[m][n]
```

**Problem:**
Write a function to count the longest repeating subsequences such that the two subsequences don't have same string characters at same positions.

assert find_longest_repeating_subseq("AABEBCDD") == 3
assert find_longest_repeating_subseq("aabb") == 2
assert find_longest_repeating_subseq("aab") == 1

**Solution:**
```
def find_longest_repeating_subseq(str):
    n = len(str)
    dp = [[0 for k in range(n+1)] for l in range(n+1)]
    for i in range(1, n+1):
        for j in range(1, n+1):
            if (str[i−1] == str[j−1] and i != j):
                dp[i][j] = 1 + dp[i−1][j−1]
            else:
                dp[i][j] = max(dp[i][j−1], dp[i−1][j])
    return dp[n][n]
```

## C  PROMPTS FOR DATA GENERATION

In this section, we present all the prompts used in our data generation process. These prompts include those for plan generation, action generation, state generation, final answer generation, semantic equivalence data generation, semantic equivalence evaluation, meta-knowledge generation, and contrastive process supervision for plan, action, and state generation.

---

**Prompt - Plan Generation**

Based on the goal, and the initial state (including the graph), propose a plan. Do not solve the problem; just outline the steps for proceeding.

Example:
### Input:

"Problem": "Solve for $a$: $\sqrt{4 + \sqrt{16 + 16a}} + \sqrt{1 + \sqrt{1 + a}} = 6$."

"Goal": "Solve $a$."

"Initial state": "We know that $\sqrt{4 + \sqrt{16 + 16a}} + \sqrt{1 + \sqrt{1 + a}} = 6$."

"Initial graph": {"Statement": {"s1": "$\sqrt{4 + \sqrt{16 + 16a}} + \sqrt{1 + \sqrt{1 + a}} = 6$"}, "Entailment": {"s1": "Given condition"}}

### Output:

"Plan": "To solve $a$, we begin by simplifying $\sqrt{4 + \sqrt{16 + 16a}}$. This simplification may also help us simplify the left side of the equation further."

---

**Prompt - Action Generation**

Based on the goal, the plan, and the history of actions and states (including graphs), propose the next action. Only specify the action itself; do not provide the outcome.

Example 1:
### Input:

"Problem": "Solve for $a$: $\sqrt{4 + \sqrt{16 + 16a}} + \sqrt{1 + \sqrt{1 + a}} = 6$."

"Goal": "Solve $a$."

"Initial state": "We know that $\sqrt{4 + \sqrt{16 + 16a}} + \sqrt{1 + \sqrt{1 + a}} = 6$."

"Initial graph": {"Statement": {"s1": "$\sqrt{4 + \sqrt{16 + 16a}} + \sqrt{1 + \sqrt{1 + a}} = 6$"}, "Entailment": {"s1": "Given condition"}}

"Plan": "To solve $a$, we begin by simplifying $\sqrt{4 + \sqrt{16 + 16a}}$. This simplification may also help us simplify the left side of the equation further."

### Output:
"Action 1": "Simplify $\sqrt{4 + \sqrt{16 + 16a}}$."

Example 2:
### Input:
"Problem": "Solve for $a$: $\sqrt{4 + \sqrt{16 + 16a}} + \sqrt{1 + \sqrt{1 + a}} = 6$."
"Goal": "Solve $a$."
"Initial state": "We know that $\sqrt{4 + \sqrt{16 + 16a}} + \sqrt{1 + \sqrt{1 + a}} = 6$."
"Initial graph": {"Statement": {"s1": "$\sqrt{4 + \sqrt{16 + 16a}} + \sqrt{1 + \sqrt{1 + a}} = 6$"}, "Entailment": {"s1": "Given condition"}}
"Plan": "To solve $a$, we begin by simplifying $\sqrt{4 + \sqrt{16 + 16a}}$. This simplification may also help us simplify the left side of the equation further."
"Action 1": "Simplify $\sqrt{4 + \sqrt{16 + 16a}}$."
"State 1": "We start by simplifying the expression under the square root.\n\nFirst, observe that:\n $16 + 16a = 16(1 + a)$\nSubstitute this back:\n$\sqrt{4 + \sqrt{16(1 + a)}}$\n\nNow, simplify $\sqrt{16(1 + a)}$:\n $\sqrt{16(1 + a)} = 4\sqrt{1 + a}$\n\nSo the expression becomes:\n$\sqrt{4 + 4\sqrt{1 + a}}$\n\nNext, factor out the 4 from inside the square root:\n$\sqrt{4(1 + \sqrt{1 + a})}$\n\nTaking the square root of 4, we get 2:\n2$\sqrt{1 + \sqrt{1 + a}}$\nSo, the simplified form of the expression is:\n2$\sqrt{1 + \sqrt{1 + a}}$."
"Graph 1": {"Statement": {"s2": "$\sqrt{4 + \sqrt{16 + 16a}} = 2\sqrt{1 + \sqrt{1 + a}}$"}, "Entailment": {"s2": "Identity equation"}}

### Output:
"Action 2": "Combine like terms $\sqrt{1 + \sqrt{1 + a}}$."

---

## Prompt - State Generation

Based on the goal, the plan, and the history of actions and states (including graphs), predict the next state. Additionally, extract the graph that illustrates key statements and their entailment relations.

Example 1:
### Input:
"Problem": "Solve for $a$: $\sqrt{4 + \sqrt{16 + 16a}} + \sqrt{1 + \sqrt{1 + a}} = 6$."
"Goal": "Solve $a$."
"Initial state": "We know that $\sqrt{4 + \sqrt{16 + 16a}} + \sqrt{1 + \sqrt{1 + a}} = 6$."
"Initial graph": {"Statement": {"s1": "$\sqrt{4 + \sqrt{16 + 16a}} + \sqrt{1 + \sqrt{1 + a}} = 6$"}, "Entailment": {"s1": "Given condition"}}
"Plan": "To solve $a$, we begin by simplifying $\sqrt{4 + \sqrt{16 + 16a}}$. This simplification may also help us simplify the left side of the equation further."
"Action 1": "Simplify $\sqrt{4 + \sqrt{16 + 16a}}$."

### Output:
"State 1": "We start by simplifying the expression under the square root.\n\nFirst, observe that:\n $16 + 16a = 16(1 + a)$\nSubstitute this back:\n$\sqrt{4 + \sqrt{16(1 + a)}}$\n\nNow, simplify $\sqrt{16(1 + a)}$:\n $\sqrt{16(1 + a)} = 4\sqrt{1 + a}$\n\nSo the expression becomes:\n$\sqrt{4 + 4\sqrt{1 + a}}$\n\nNext, factor out the 4 from inside the square root:\n$\sqrt{4(1 + \sqrt{1 + a})}$\n\nTaking the square root of 4, we get 2:\n2$\sqrt{1 + \sqrt{1 + a}}$\nSo, the simplified form of the expression is:\n2$\sqrt{1 + \sqrt{1 + a}}$."
"Graph 1": {"Statement": {"s2": "$\sqrt{4 + \sqrt{16 + 16a}} = 2\sqrt{1 + \sqrt{1 + a}}$"}, "Entailment": {"s2": "Identity equation"}}

Example 2:
### Input:
"Problem": "Solve for $a$: $\sqrt{4 + \sqrt{16 + 16a}} + \sqrt{1 + \sqrt{1 + a}} = 6$."
"Goal": "Solve $a$."

"Initial state": "We know that $\sqrt{4 + \sqrt{16 + 16a}} + \sqrt{1 + \sqrt{1 + a}} = 6$."

"Initial graph": {"Statement": {"s1": "$\sqrt{4 + \sqrt{16 + 16a}} + \sqrt{1 + \sqrt{1 + a}} = 6$"}, "Entailment": {"s1": "Given condition"}}

"Plan": "To solve $a$, we begin by simplifying $\sqrt{4 + \sqrt{16 + 16a}}$. This simplification may also help us simplify the left side of the equation further."

"Action 1": "Simplify $\sqrt{4 + \sqrt{16 + 16a}}$."

"State 1": "We start by simplifying the expression under the square root.\n\nFirst, observe that:\n $16 + 16a = 16(1 + a)$\nSubstitute this back:\n$\sqrt{4 + \sqrt{16(1 + a)}}$\n\nNow, simplify $\sqrt{16(1 + a)}$:\n $\sqrt{16(1 + a)} = 4\sqrt{1 + a}$\n\nSo the expression becomes:\n$\sqrt{4 + 4\sqrt{1 + a}}$\n\nNext, factor out the 4 from inside the square root:\n$\sqrt{4(1 + \sqrt{1 + a})}$\n\nTaking the square root of 4, we get 2:\n$2\sqrt{1 + \sqrt{1 + a}}$\nSo, the simplified form of the expression is:\n$2\sqrt{1 + \sqrt{1 + a}}$."

"Graph 1": {"Statement": {"s2": "$\sqrt{4 + \sqrt{16 + 16a}} = 2\sqrt{1 + \sqrt{1 + a}}$"}, "Entailment": {"s2": "Identity equation"}}

"Action 2": "Combine like terms $\sqrt{1 + \sqrt{1 + a}}$."

### Output:
"State 2": "We can obtain $\sqrt{4 + \sqrt{16 + 16a}} + \sqrt{1 + \sqrt{1 + a}} = 2\sqrt{1 + \sqrt{1 + a}} + \sqrt{1 + \sqrt{1 + a}} = 3\sqrt{1 + \sqrt{1 + a}}$."

"Graph 2": {"Statement": {"s3": "$\sqrt{4 + \sqrt{16 + 16a}} + \sqrt{1 + \sqrt{1 + a}} = 3\sqrt{1 + \sqrt{1 + a}}$"}, "Entailment": {"s3": ["s1", "s2"]}}

---

## Prompt - Final Answer Generation

Based on the goal and the current state (including the graph), determine if the goal has been achieved. If it has, generate the final answer; otherwise, return "Not yet".

Example 1:
### Input:
"Problem": "Solve for $a$: $\sqrt{4 + \sqrt{16 + 16a}} + \sqrt{1 + \sqrt{1 + a}} = 6$."

"Goal": "Solve $a$."

"Initial state": "We know that $\sqrt{4 + \sqrt{16 + 16a}} + \sqrt{1 + \sqrt{1 + a}} = 6$."

"Initial graph": {"Statement": {"s1": "$\sqrt{4 + \sqrt{16 + 16a}} + \sqrt{1 + \sqrt{1 + a}} = 6$"}, "Entailment": {"s1": "Given condition"}}

"Plan": "To solve $a$, we begin by simplifying $\sqrt{4 + \sqrt{16 + 16a}}$. This simplification may also help us simplify the left side of the equation further."

"Action 1": "Simplify $\sqrt{4 + \sqrt{16 + 16a}}$."

"State 1": "We start by simplifying the expression under the square root.\n\nFirst, observe that:\n $16 + 16a = 16(1 + a)$\nSubstitute this back:\n$\sqrt{4 + \sqrt{16(1 + a)}}$\n\nNow, simplify $\sqrt{16(1 + a)}$:\n $\sqrt{16(1 + a)} = 4\sqrt{1 + a}$\n\nSo the expression becomes:\n$\sqrt{4 + 4\sqrt{1 + a}}$\n\nNext, factor out the 4 from inside the square root:\n$\sqrt{4(1 + \sqrt{1 + a})}$\n\nTaking the square root of 4, we get 2:\n$2\sqrt{1 + \sqrt{1 + a}}$\nSo, the simplified form of the expression is:\n$2\sqrt{1 + \sqrt{1 + a}}$."

"Graph 1": {"Statement": {"s2": "$\sqrt{4 + \sqrt{16 + 16a}} = 2\sqrt{1 + \sqrt{1 + a}}$"}, "Entailment": {"s2": "Identity equation"}}

"Action 2": "Combine like terms $\sqrt{1 + \sqrt{1 + a}}$."

"State 2": "We can obtain $\sqrt{4 + \sqrt{16 + 16a}} + \sqrt{1 + \sqrt{1 + a}} = 2\sqrt{1 + \sqrt{1 + a}} + \sqrt{1 + \sqrt{1 + a}} = 3\sqrt{1 + \sqrt{1 + a}}$."

"Graph 2": {"Statement": {"s3": "$\sqrt{4 + \sqrt{16 + 16a}} + \sqrt{1 + \sqrt{1 + a}} = 3\sqrt{1 + \sqrt{1 + a}}$"}, "Entailment": {"s3": ["s1", "s2"]}}

### Output:
"Not yet"

Example 2:

### Input:
"Problem": "Solve for $a$: $\sqrt{4 + \sqrt{16 + 16a}} + \sqrt{1 + \sqrt{1 + a}} = 6$."
"Goal": "Solve $a$."
"Initial state": "We know that $\sqrt{4 + \sqrt{16 + 16a}} + \sqrt{1 + \sqrt{1 + a}} = 6$."
"Initial graph": {"Statement": {"s1": "$\sqrt{4 + \sqrt{16 + 16a}} + \sqrt{1 + \sqrt{1 + a}} = 6$"}, "Entailment": {"s1": "Given condition"}}
"Plan": "To solve $a$, we begin by simplifying $\sqrt{4 + \sqrt{16 + 16a}}$. This simplification may also help us simplify the left side of the equation further."
"Action 1": "Simplify $\sqrt{4 + \sqrt{16 + 16a}}$."
"State 1": "We start by simplifying the expression under the square root.\n\nFirst, observe that:\n $16 + 16a = 16(1 + a)$\nSubstitute this back:\n$\sqrt{4 + \sqrt{16(1 + a)}}$\n\nNow, simplify $\sqrt{16(1 + a)}$:\n $\sqrt{16(1 + a)} = 4\sqrt{1 + a}$\n\nSo the expression becomes:\n$\sqrt{4 + 4\sqrt{1 + a}}$\n\nNext, factor out the 4 from inside the square root:\n$\sqrt{4(1 + \sqrt{1 + a})}$\n\nTaking the square root of 4, we get 2:\n$2\sqrt{1 + \sqrt{1 + a}}$\nSo, the simplified form of the expression is:\n$2\sqrt{1 + \sqrt{1 + a}}$."
"Graph 1": {"Statement": {"s2": "$\sqrt{4 + \sqrt{16 + 16a}} = 2\sqrt{1 + \sqrt{1 + a}}$"}, "Entailment": {"s2": "Identity equation"}}
"Action 2": "Combine like terms $\sqrt{1 + \sqrt{1 + a}}$."
"State 2": "We can obtain $\sqrt{4 + \sqrt{16 + 16a}} + \sqrt{1 + \sqrt{1 + a}} = 2\sqrt{1 + \sqrt{1 + a}} + \sqrt{1 + \sqrt{1 + a}} = 3\sqrt{1 + \sqrt{1 + a}}$."
"Graph 2": {"Statement": {"s3": "$\sqrt{4 + \sqrt{16 + 16a}} + \sqrt{1 + \sqrt{1 + a}} = 3\sqrt{1 + \sqrt{1 + a}}$"}, "Entailment": {"s3": ["s1", "s2"]}}
"Action 3": "Solve $a$."
"State 3": "Isolate the square root term by dividing both sides by 3:\n$\sqrt{1 + \sqrt{1 + a}} = 2$\n\nSquare both sides: $\left(\sqrt{1 + \sqrt{1 + a}}\right)^2 = 2^2$\n$1 + \sqrt{1 + a} = 4$\n\nIsolate the inner square root:\n$\sqrt{1 + a} = 4 - 1$\n$\sqrt{1 + a} = 3$\n\nSquare both sides again:\n$\left(\sqrt{1 + a}\right)^2 = 3^2$\n$1 + a = 9$\n\nSolve for $a$:\n$a = 9 - 1$\n$a = 8$\n\nThe solution is $a = 8$."
"Graph 3": {"Statement": {"s4": "$\sqrt{1 + \sqrt{1 + a}} = 2$", "s5": "$a = 8$"}, "Entailment": {"s4": ["s1", "s3"], "s5": ["s4"]}}

### Output:
"Final answer": "8"

---

## Prompt - Semantic Equivalence Data Generation

Rewrite the given sentence into two or three different versions. Ensure that each version is distinct in wording and structure. Provide both a thought process and a final answer. In the thought process, include as many details as possible, ensuring that no steps are omitted.

Example 1:
### Input:
"Sentence": "$\frac{XZ}{XY} = \frac{ZY}{XY} = \frac{1}{2}$."

### Output:
"Thought": "There are various ways to rewrite the given sentence: 1. $\frac{XZ}{XY} = \frac{1}{2}$ and $\frac{ZY}{XY} = \frac{1}{2}$. 2. $XZ = ZY = \frac{1}{2}XY$. These paraphrased versions capture the same relationship as the original equation.",
"Answer": ["$\frac{XZ}{XY} = \frac{1}{2}$ and $\frac{ZY}{XY} = \frac{1}{2}$.", "$XZ = ZY = \frac{1}{2}XY$."]

Example 2:
### Input:
"Sentence": "$125a$ is a three−digit number."

### Output:
"Thought": "To rewrite the sentence '$125a$ is a three−digit number', we need to express it in different forms that convey the same meaning. The given sentence implies that $125a$ is within the range of three−digit numbers, i.e., between 100 and 999. Let's break down and reframe the statement as follows: 1. $100 \leq 125a \leq 999$. 2. 'The value of $125a$ lies within the interval [100, 999]'. 3. '$125a$ is an integer that

satisfies the condition $100 \leq 125a \leq 999$'. Each of these reframed versions captures the same relationship as the original sentence."

"Answer": ["$100 \leq 125a \leq 999$.", "The value of $125a$ lies within the interval [100, 999]", "$125a$ is an integer that satisfies the condition $100 \leq 125a \leq 999$."]

---

### Prompt - Semantic Equivalence Evaluation

Compare the provided candidate options, considering both their current attributes and potential future outcomes (if applicable). Determine whether they are semantically equivalent, and respond with either "same" or "different".

Example:
### Input:
"Problem": "Given the polynomials $p(x) = 2x + 3$ and $q(x) = x^2 - x + 4$, find the polynomial resulting from multiplying $p(x)$ and $q(x)$ and express it in standard polynomial form. What is the coefficient of $x^2$ in the resulting polynomial?"
"Goal": "Find the coefficient of $x^2$ in the resulting polynomial from multiplying $p(x)$ and $q(x)$.",
"Initial state": "We have the polynomials $p(x) = 2x + 3$ and $q(x) = x^2 - x + 4$.",
"Initial graph": {"Statement": {"s1": "$p(x) = 2x + 3$", "s2": "$q(x) = x^2 - x + 4$"}, "Entailment": {"s1": "Given condition", "s2": "Given condition"}},
"Plan": "First, multiply the polynomials $p(x)$ and $q(x)$. Then, identify the coefficient of $x^2$ in the resulting polynomial.",
"Action 1": "Multiply the polynomials $p(x)$ and $q(x)$.",
"State 1": "To multiply $p(x) = 2x + 3$ and $q(x) = x^2 - x + 4$, distribute each term of $p(x)$ to each term of $q(x)$: $(2x + 3)(x^2 - x + 4) = 2x(x^2) + 2x(-x) + 2x(4) + 3(x^2) + 3(-x) + 3(4) = 2x^3 - 2x^2 + 8x + 3x^2 - 3x + 12$.",
"Graph 1": {"Statement": {"s3": "$p(x)q(x) = 2x^3 - 2x^2 + 8x + 3x^2 - 3x + 12$"}, "Entailment": {"s3": ["s1", "s2"]}},
"Action 2": "Combine like terms in the resulting polynomial.",
"State 2": {
    "Option 1": "$2x^3 + (2x^2 - 2x^2) + 8x + 3x^2 + (-3x) + 12 = 2x^3 + 3x^2 + 5x + 12$",
    "Option 2": "$2x^3 + (-2x^2 + 3x^2) + (8x - 3x) + 12 = 2x^3 + x^2 + 5x + 12$",
}
"Future": {
    "Future 1": {"Graph 2": {"Statement": {"s4": "$p(x)q(x) = 2x^3 + 3x^2 + 5x + 12$"}, "Entailment": {"s4": ["s3"]}}},
    "Future 2": {"Graph 2": {"Statement": {"s4": "$p(x)q(x) = 2x^3 + x^2 + 5x + 12$"}, "Entailment": {"s4": ["s3"]}}}
}

### Output:
"Different"

---

### Prompt - Meta Knowledge Generation

Provide the general knowledge needed to solve the problem. Do not provide a specific solution plan. Instead, offer knowledge that can help verify answers and identify errors.

Example 1:
### Input:
"Problem": "Consider the quadratic equation $ax^2 + bx + c = 0$ where $a$, $b$, and $c$ are integers. For this equation to have integer solutions, the discriminant $b^2 - 4ac$ must be a perfect square. Suppose the quadratic equation $2x^2 + kx + 3 = 0$ has integer solutions. What is the value of $k$?"

### Output:
"General knowledge": "To avoid errors when determining the value of variables in algebraic equations, follow these general guidelines: 1. Understand conditions for solutions: Know the conditions required for specific solution types and how they relate to equation parameters. 2. Discriminant analysis: Use the discriminant to determine solution types and ensure it meets conditions for the desired type. 3. Factorization techniques: Use factorization to simplify equations and find relationships between parameters. 4. Systematic approach to

finding solutions: Solve related equations systematically and consider all potential factor pairs. 5. Verification: Verify solutions by substituting back to confirm correctness and catch errors."

Example 2:
### Input:
"Problem": "Given the polynomials $p(x) = 2x + 3$ and $q(x) = x^2 - x + 4$, find the polynomial resulting from multiplying $p(x)$ and $q(x)$ and express it in standard polynomial form. What is the coefficient of $x^2$ in the resulting polynomial?"

### Output:
"General knowledge": "To avoid mistakes when solving polynomial multiplication problems, follow these general guidelines: 1. Understand Polynomial Terms and Their Degrees: Recognize the individual terms of a polynomial and their respective degrees. 2. Distributive Property in Polynomial Multiplication: Apply the distributive property correctly by multiplying each term of the first polynomial by each term of the second polynomial. 3. Combine Like Terms: After distributing, combine the like terms, which are terms with the same degree. Be systematic in organizing terms to ensure all like terms are combined correctly. 4. Pay Attention to Signs: Be careful with positive and negative signs during multiplication and when combining like terms. Ensure that the signs of the terms are handled correctly during the distribution process."

---

## Prompt - Contrastive Process Supervision for Plan Generation

Compare the provided candidate options, considering both their current attributes and potential future outcomes (if applicable). Pay attention to the meta knowledge. First, present a detailed comparison, showing every step without skipping any. Then, provide a conclusion, selecting only one answer.

Example:
### Input:
"Meta knowledge": "To avoid errors when determining the value of variables in algebraic equations, follow these general guidelines: 1. Understand conditions for solutions: Know the conditions required for specific solution types and how they relate to equation parameters. 2. Discriminant analysis: Use the discriminant to determine solution types and ensure it meets conditions for the desired type. 3. Factorization techniques: Use factorization to simplify equations and find relationships between parameters. 4. Systematic approach to finding solutions: Solve related equations systematically and consider all potential factor pairs. 5. Verification: Verify solutions by substituting back to confirm correctness and catch errors.",
"Problem": "Consider the quadratic equation $ax^2 + bx + c = 0$ where $a$, $b$, and $c$ are integers. For this equation to have integer solutions, the discriminant $b^2 - 4ac$ must be a perfect square. Suppose the quadratic equation $2x^2 + kx + 3 = 0$ has integer solutions. What is the value of $k$?",
"Goal": "Find the value of $k$.",
"Initial state": "We know that $2x^2 + kx + 3 = 0$ has integer solutions.",
"Initial graph": {"Statement": {"s1": "$2x^2 + kx + 3 = 0$ has integer solutions."}, "Entailment": {"s1": "Given condition"}},
"Plan": {
    "Option 1": "We can use the fact that the discriminant of a quadratic equation must be a perfect square to write an equation in terms of $k$ and then solve for $k$.",
    "Option 2": "The discriminant $b^2 - 4ac$ must be a perfect square. We will set up the discriminant and solve for $k$ under the condition.",
    "Option 3": "We know that the discriminant $b^2 - 4ac$ must be a perfect square. Let's try to find the value of $k$ for which the discriminant is a perfect square."
},
"Future": {
    "Future 1": {"Action 1": "Find the discriminant.", "State 1": "The discriminant is $b^2 - 4ac$, so in this case, it is $k^2 - 24$.", "Graph 1": {"Statement": {"s2": "The discriminant is $k^2 - 24$."}, "Entailment": {"s2": ["s1"]}}, "Action 2": "Since the discriminant must be a perfect square, we can set $k^2 - 24 = x^2$ for some integer $x$.", "State 2": "Now, we need to find $k$ by solving the equation $k^2 - 24 = x^2$. We can do this by completing the square: $k^2 - 24 = x^2$. Add 24 to both sides: $k^2 = x^2 + 24$. Take the square root of both sides: $k = \pm\sqrt{x^2 + 24}$.", "Graph 2": {"Statement": {"s3": "$k = \pm\sqrt{x^2 + 24}$"}, "Entailment": {"s3": ["s1", "s2"]}}, "Action 3": "Since $k$ is an integer, $x^2 + 24$ must also be a perfect square. Let $x^2 + 24 = y^2$ for some integer $y$. Then, $k = \pm y$.", "State 3": "The only values of $y$ that work are $y = \pm 4$, so $k = \pm 4$. The final answer is $k = 4$ or $k = -4$.", "Graph 3": {"Statement": {"s4": "$k = \pm 4$"}, "Entailment": {"s4": ["s3"]}}},
    "Future 2": {"Action 1": "Calculate the discriminant $b^2 - 4ac$ for the quadratic equation.", "State 1": "The discriminant $\Delta$ for the equation is given by: $\Delta = k^2 - 4 \cdot 2 \cdot 3 = k^2 - 24$.", "Graph 1": {"Statement": {"s2":

"$\Delta = k^2 - 24$.", "Entailment": {"s2": ["s1"]}}, "Action 2": "Set the discriminant $k^2 - 24$ to be a perfect square.", "State 2": "For $k^2 - 24$ to be a perfect square, let $k^2 - 24 = n^2$ for some integer $n$. This gives us the equation: $k^2 - n^2 = 24$ $(k - n)(k + n) = 24$.", "Graph 2": {"Statement": {"s3": "$(k - n)(k + n) = 24$"}, "Entailment": {"s3": ["s1", "s2"]}}, "Action 3": "Solve for $k$ by finding pairs $(k - n)$ and $(k + n)$ that multiply to 24.", "State 3": "We need to find integer pairs $(k - n)$ and $(k + n)$ such that their product is 24. Possible pairs are: $(1, 24), (2, 12), (3, 8), (4, 6), (-1, -24), (-2, -12), (-3, -8), (-4, -6)$. Solving for $k$ in each case, we find: $k - n = 2, k + n = 12, 2k = 14, k = 7$.", "Graph 3": {"Statement": {"s4": "$k = 7$"}, "Entailment": {"s4": ["s3"]}}},

"Future 3": {"Action 1": "Find the discriminant of the equation.", "State 1": "The discriminant is $b^2 - 4ac$ = $k^2 - 4(2)(3) = k^2 - 24$.", "Graph 1": {"Statement": {"s2": "The discriminant is $k^2 - 24$."}, "Entailment": {"s2": ["s1"]}}, "Action 2": "Make the discriminant a perfect square.", "State 2": "For the discriminant $k^2 - 24$ to be a perfect square, $k^2$ must be a perfect square plus 24. Let's call $k^2 = m^2$ for some integer $m$. Then, $m^2 - 24$ must be a perfect square. This means that $m^2$ must be 25, 49, 64, 81, or 100, since these are the perfect squares closest to 24.", "Graph 2": {"Statement": {"s3": "$k^2$ must be 25, 49, 64, 81, or 100."}, "Entailment": {"s3": ["s1", "s2"]}}, "Action 3": "Find the corresponding values of $k$.", "State 3": "If $m^2 = 25$, then $m = 5$, and $k = 5$. If $m^2 = 49$, then $m = 7$, and $k = 7$. If $m^2 = 64$, then $m = 8$, and $k = 8$. If $m^2 = 81$, then $m = 9$, and $k = 9$. If $m^2 = 100$, then $m = 10$, and $k = 10$.", "Graph 3": {"Statement": {"s4": "$k = 5, 7, 8, 9, 10$"}, "Entailment": {"s4": ["s3"]}}}}
}

### Output:
"Comparison": "All options mention that the discriminant must be a perfect square. Based on this observation, they will solve for $k$. All candidates are valid. However, the futures are different. In Future 1, $k^2 = x^2 + 24$ leads to $k = 4$ or $k = -4$, which is incorrect. Future 2 is correct. In Future 3, it mentions $m^2 - 24$ must be a perfect square, then $m^2$ must be 25, 49, 64, 81, or 100, which is incorrect.",
"Conclusion": "Option 2"

---

## Prompt - Contrastive Process Supervision for Action Generation

Compare the provided candidate options, considering both their current attributes and potential future outcomes (if applicable). Pay attention to the meta knowledge. First, present a detailed comparison, showing every step without skipping any. Then, provide a conclusion, selecting only one answer.

Example:
### Input:
"Meta knowledge": "To avoid errors when determining the value of variables in algebraic equations, follow these general guidelines: 1. Understand conditions for solutions: Know the conditions required for specific solution types and how they relate to equation parameters. 2. Discriminant analysis: Use the discriminant to determine solution types and ensure it meets conditions for the desired type. 3. Factorization techniques: Use factorization to simplify equations and find relationships between parameters. 4. Systematic approach to finding solutions: Solve related equations systematically and consider all potential factor pairs. 5. Verification: Verify solutions by substituting back to confirm correctness and catch errors.",
"Problem": "Consider the quadratic equation $ax^2 + bx + c = 0$ where $a$, $b$, and $c$ are integers. For this equation to have integer solutions, the discriminant $b^2 - 4ac$ must be a perfect square. Suppose the quadratic equation $2x^2 + kx + 3 = 0$ has integer solutions. What is the value of $k$?",
"Goal": "Find the value of $k$.",
"Initial state": "We know that $2x^2 + kx + 3 = 0$ has integer solutions.",
"Initial graph": {"Statement": {"s1": "$2x^2 + kx + 3 = 0$ has integer solutions."}, "Entailment": {"s1": "Given condition"}},
"Plan": "The discriminant $b^2 - 4ac$ must be a perfect square. We will set up the discriminant and solve for $k$ under the condition.",
"Action 1": "Calculate the discriminant $b^2 - 4ac$ for the quadratic equation.", "State 1": "The discriminant $\Delta$ for the equation is given by: $\Delta = k^2 - 4 \cdot 2 \cdot 3 = k^2 - 24$.",
"Graph 1": {"Statement": {"s2": "$\Delta = k^2 - 24$.", "Entailment": {"s2": ["s1"]}},
"Action 2": "Set the discriminant $k^2 - 24$ to be a perfect square.",
"State 2": "For $k^2 - 24$ to be a perfect square, let $k^2 - 24 = n^2$ for some integer $n$. This gives us the equation: $k^2 - n^2 = 24$ $(k - n)(k + n) = 24$.",
"Graph 2": {"Statement": {"s3": "$(k - n)(k + n) = 24$"}, "Entailment": {"s3": ["s1", "s2"]}},
"Action 3": {
  "Option 1": "Solve for $k$ by finding pairs $(k - n)$ and $(k + n)$ that multiply to 24.",

"Option 2": "Factor the right side of the equation: $(k - n)(k + n) = 24$.",
    "Option 3": "Solve for $k$ by factoring $(k - n)(k + n) = 24$."
},
"Future": {
    "Future 1": {"State 3": "We need to find integer pairs $(k - n)$ and $(k + n)$ such that their product is 24. Possible pairs are: $(1, 24), (2, 12), (3, 8), (4, 6), (-1, -24), (-2, -12), (-3, -8), (-4, -6)$. Solving for $k$ in each case, we find: $k - n = 2, k + n = 12, 2k = 14, k = 7$.", "Graph 3": {"Statement": {"s4": "$k = 7$"}, "Entailment": {"s4": ["s3"]}}},
    "Future 2": {"State 3": "The factors of 24 are 1, 2, 3, 4, 6, 8, 12, and 24. We can check each of these to see if they can be expressed as the product of two integers with a sum of $k$ and a difference of $n$. $1 \cdot 24 = 24$, $2 \cdot 12 = 24$, and $3 \cdot 8 = 24$ are the only possibilities. Now, let's check these: $k - n = 1, k + n = 24$, $k = 12, n = 23, k - n = 2, k + n = 12, k = 7, n = 5$, and $k - n = 3, k + n = 8, k = 5, n = 5$. So, the possible values of $k$ are 5, 7, and 12. However, we need to check if $k$ is an integer.", "Graph 3": {"Statement": {"s4": "$k = 5, 7, 12$"}, "Entailment": {"s4": ["s3"]}}},
    "Future 3": {"State 3": "We need to find pairs of integers $(k - n)$ and $(k + n)$ that multiply to 24. The possible pairs are $(1, 24), (2, 12), (3, 8), (4, 6), (-1, -24), (-2, -12), (-3, -8), (-4, -6)$. For each pair, solve for $k$ and $n$.", "Graph 3": {"Statement": {"s4": "Possible pairs for $(k - n)$ and $(k + n)$ are $(1, 24)$, $(2, 12), (3, 8), (4, 6), (-1, -24), (-2, -12), (-3, -8), (-4, -6)$."}, "Entailment": {"s4": ["s3"]}}}
}

### Output:
"Comparison": "All options involve solving for $k$ based on the equation. Option 1 mentions finding pairs. Option 2 and 3 mention factorization. All candidates are valid. Given the actions, the future states are different. Future 1 shows $k = 7$. Future 2 shows $k = 5, 7, 12$. Future 3 does not mention $k$. Future 1 is correct. In Future 2, by solving $k - n = 1, k + n = 24$, we obtain $k = 12.5, n = 11.5$ rather than $k = 12, n = 23$. Thus, Future 2 is incorrect. Future 3 does not mention $k$.",
"Conclusion": "Option 1"

---

## Prompt - Contrastive Process Supervision for State Generation

Compare the provided candidate options, considering both their current attributes and potential future outcomes (if applicable). Pay attention to the meta knowledge. First, present a detailed comparison, showing every step without skipping any. Then, provide a conclusion, selecting only one answer.

Example:
### Input:
"Meta knowledge": "To avoid mistakes when solving polynomial multiplication problems, follow these general guidelines: 1. Understand Polynomial Terms and Their Degrees: Recognize the individual terms of a polynomial and their respective degrees. 2. Distributive Property in Polynomial Multiplication: Apply the distributive property correctly by multiplying each term of the first polynomial by each term of the second polynomial. 3. Combine Like Terms: After distributing, combine the like terms, which are terms with the same degree. Be systematic in organizing terms to ensure all like terms are combined correctly. 4. Pay Attention to Signs: Be careful with positive and negative signs during multiplication and when combining like terms. Ensure that the signs of the terms are handled correctly during the distribution process.",
"Problem": "Given the polynomials $p(x) = 2x + 3$ and $q(x) = x^2 - x + 4$, find the polynomial resulting from multiplying $p(x)$ and $q(x)$ and express it in standard polynomial form. What is the coefficient of $x^2$ in the resulting polynomial?",
"Goal": "Find the coefficient of $x^2$ in the resulting polynomial from multiplying $p(x)$ and $q(x)$.",
"Initial state": "We have the polynomials $p(x) = 2x + 3$ and $q(x) = x^2 - x + 4$.",
"Initial graph": {"Statement": {"s1": "$p(x) = 2x + 3$", "s2": "$q(x) = x^2 - x + 4$"}, "Entailment": {"s1": "Given condition", "s2": "Given condition"}},
"Plan": "First, multiply the polynomials $p(x)$ and $q(x)$. Then, identify the coefficient of $x^2$ in the resulting polynomial.",
"Action 1": "Multiply the polynomials $p(x)$ and $q(x)$.",
"State 1": "To multiply $p(x) = 2x + 3$ and $q(x) = x^2 - x + 4$, distribute each term of $p(x)$ to each term of $q(x)$: $(2x + 3)(x^2 - x + 4) = 2x(x^2) + 2x(-x) + 2x(4) + 3(x^2) + 3(-x) + 3(4) = 2x^3 - 2x^2 + 8x + 3x^2 - 3x + 12$.",
"Graph 1": {"Statement": {"s3": "$p(x)q(x) = 2x^3 - 2x^2 + 8x + 3x^2 - 3x + 12$"}, "Entailment": {"s3": ["s1", "s2"]}},
"Action 2": "Combine like terms in the resulting polynomial.",
"State 2": {
    "Option 1": "$2x^3 + (2x^2 - 2x^2) + 8x + 3x^2 + (-3x) + 12 = 2x^3 + 3x^2 + 5x + 12$",

"Option 2": "$2x^3 + (-2x^2 + 3x^2) + (8x - 3x) + 12 = 2x^3 + x^2 + 5x + 12$",
        "Option 3": "$2x^3 - 2x^2 + 8x + 3x^2 + 12 = 2x^3 + x^2 + 8x + 12$"
},
"Future": {
        "Future 1": {"Graph 2": {"Statement": {"s4": "$p(x)q(x) = 2x^3 + 3x^2 + 5x + 12$"}, "Entailment": {"s4":
["s3"]}}},
        "Future 2": {"Graph 2": {"Statement": {"s4": "$p(x)q(x) = 2x^3 + x^2 + 5x + 12$"}, "Entailment": {"s4":
["s3"]}}}
        "Future 3": {"Graph 2": {"Statement": {"s4": "$p(x)q(x) = 2x^3 + x^2 + 8x + 12$"}, "Entailment": {"s4":
["s3"]}}}
}

### Output:
"Comparison":
"All options involve combining like terms to find the coefficient of $x^2$. Option 1 combines terms resulting in
$2x^3 + 3x^2 + 5x + 12$, Option 2 simplifies to $2x^3 + x^2 + 5x + 12$, and Option 3 simplifies to $2x^3 + x^2 +$
$8x + 12$. Thus, they yield different coefficients for $x^2$ and $x$. Option 1 mentions $2x^3 + (2x^2 - 2x^2) + 8x +$
$3x^2 + (-3x) + 12$. However, State 1 gives $2x^3 - 2x^2 + 8x + 3x^2 - 3x + 12$. Option 1 mistakes $-2x^2$ as
$2x^2 - 2x^2$. On the other hand, Option 2 uses the correct equation $2x^3 + (-2x^2 + 3x^2) + (8x - 3x) + 12$,
which leads to the correct results. The problem of Option 3 is that it omits the term of $-3x$, leading to a
wrong coefficient for $x$.",
"Conclusion": "Option 2"

## D  IMPLEMENTATION DETAILS

SWAP is fine-tuned from a base language model using LoRA. To enable scalability and general-
ization in our framework, we fine-tune a single generator and a single discriminator, and repurpose
them to serve as the policy model, world model, and controller. For each dataset, the generator
is fine-tuned on all positive trajectories in the training set that lead to the correct final answer. As
illustrated in Figure 2, the generator contains two LoRAs. The original LoRA is fine-tuned on the
positive trajectories as usual, while the SemEquiv-LoRA is fine-tuned on semantic equivalence data,
which are bootstrapped using GPT-4o, for plan, actions and states. Specially, the number of tra-
jectories for the generator are as follows: GSM8k (28.3k), MATH (49.3k), ReClor (14.5k), FOLIO
(7.3k), HumanEval (3.1k), and MBPP (1.3k). For each positive trajectory, we random sampled some
steps and generated two alternatives for each step. The number of semantically equivalent pairs we
obtained are as follows: GSM8k (8.1k), MATH (24.2k), ReClor (7.1k), FOLIO (3.8k), HumanEval
(1.6k) and MBPP (0.7k).

The discriminator is fine-tuned on contrastive process annotations for every dataset (Figure 3).
Specifically, given a positive trajectory, we randomly search two alternatives for each step and ob-
tain their ranking. The number of trajectories for the discriminator are as follows: GSM8k (48.0k),
MATH (98.2k), ReClor (28.7k), FOLIO (14.1k), HumanEval (6.0k), and MBPP (2.5k). We boot-
strap the meta-knowledge text for each training question using GPT-4o. For inference, we use a
DPR model (Karpukhin et al., 2020) to obtain embeddings for both training questions and the test
query, then calculate the Cosine similarity to select the top 5 matches. Once the relevant knowledge
is extracted from these top 5 training questions, we consider two approaches: 1) using the original
text directly or 2) compressing it into a shorter version. In our experiments, we found that Approach
1 resulted in higher accuracy, whereas Approach 2 offered slightly lower accuracy but faster infer-
ence speed. The length of the future trajectory $\tau^j$ is also determined experimentally. We found that
the optimal strategy for plan discrimination is to include all future steps. For action discrimination,
including only the next state led to an increase in discrimination accuracy, whereas including ad-
ditional future steps caused the accuracy to decrease. We attribute this decline to the new errors
introduced by the subsequent steps after analyzing the error cases. Additionally, we observed a class
imbalance issue in the contrastive process annotations. Specifically, when generating discrimination
data (for both actions and states), GPT-4o tends to select the first candidate if all options are similar.
To address this, we propose two strategies: 1) Pre-processing: During data generation, we randomly
alter the index of the ground truth. For samples where GPT-4o cannot provide the correct answer,
we supply the ground truth to assist the model. 2) Post-processing: After generating the training

Table 3: Fine-grained performance on MATH across different subsets: Algebra (ALG), Counting and Probability (CP), Geometry (GEO), Intermediate Algebra (IA), Number Theory (NT), Precalculus (PRE), and Prealgebra (PALG). The number of test questions for each subset is shown in parentheses. Bold values indicate the best performance per subset and overall.

| Model | Math | | | | | | | |
|---|---|---|---|---|---|---|---|---|
| | ALG (1187) | CP (474) | GEO (479) | IA (903) | NT (540) | PRE (546) | PALG (871) | Total (5000) |
| LLaMA3-8B (0-shot CoT) | 38.6 | 21.5 | 20.0 | 12.2 | 21.3 | 15.2 | 47.5 | 27.6 |
| LLaMA3-8B (4-shot CoT) | 35.0 | 19.4 | 17.7 | 8.1 | 17.2 | 11.9 | 41.2 | 23.6 |
| LLaMA3-8B (0-shot CoT + SC) | 38.4 | 19.9 | 17.3 | 10.4 | 18.3 | 12.6 | 46.4 | 26.0 |
| LLaMA3-8B (SFT-CoT) | 37.5 | 19.7 | 18.0 | 10.1 | 18.5 | 11.7 | 45.0 | 25.4 |
| SWAP (w/o discriminator) | 51.0 | 40.2 | 27.4 | 18.0 | 30.9 | 18.2 | 60.7 | 37.3 |
| SWAP | **55.4** | **43.4** | **31.8** | **22.3** | **37.8** | **23.1** | **68.3** | **42.3** |

Table 4: Fine-grained performance on MATH across different difficulty levels (Level 1-5). The number of test questions for each level is shown. Bold values indicate the best performance.

| Model | Math | | | | | |
|---|---|---|---|---|---|---|
| | L1 (437) | L2 (894) | L3 (1131) | L4 (1214) | L5 (1324) | Total (5000) |
| LLaMA3-8B (0-shot CoT) | 65.2 | 45.4 | 30.7 | 18.9 | 8.4 | 27.6 |
| LLaMA3-8B (4-shot CoT) | 54.7 | 38.3 | 27.1 | 16.5 | 7.2 | 23.6 |
| LLaMA3-8B (0-shot CoT + SC) | 65.2 | 45.0 | 29.3 | 16.4 | 6.2 | 26.0 |
| LLaMA3-8B (SFT-CoT) | 64.4 | 44.5 | 28.7 | 15.9 | 5.7 | 25.4 |
| SWAP (w/o discriminator) | 76.5 | 59.9 | 43.3 | 29.4 | 11.7 | 37.3 |
| SWAP | **78.8** | **65.4** | **50.6** | **34.2** | **14.9** | **42.3** |

data, we manually change the index of the options and adjust the output accordingly. To further enhance model robustness, we also apply data augmentation by increasing the training data through varying the index and description of the options.

To ensure effective training, we also employ specialized strategies such as curriculum learning and self-improving training. For curriculum learning, we first divide the training questions into groups based on their difficulty levels. For some datasets, such as MATH, the difficulty level of the problems is already provided; for other datasets, we determine the difficulty level based on the length of the solution. In the first round, we use Level 1 problems; in the second round, use both Level 1 and Level 2 problems, and so on. In each round, we train the model until convergence, using early stopping to prevent overfitting. We also employ self-improving training to iteratively refine the model's accuracy. After training, the system is run on the training samples, and the errors it produces are collected. These errors are then used to further fine-tune the discriminator, while the generator remains fixed. This process is repeated until convergence.

# E  FINE-GRAINED RESULTS

To gain a comprehensive understanding of the model's strengths and weaknesses, we provide fine-grained results on MATH (Table 3 and 4). We choose MATH for this analysis since it categorizes the test set by both problem types and difficulty levels, facilitating a more detailed evaluation of model performance across different dimensions. From Table 3, we observe that SWAP consistently outperforms other models across all subsets and overall, surpassing various reasoning methods applied to LLaMA3-8B-Instruct. This demonstrates that SWAP significantly enhances the overall mathematical reasoning capability compared to the baseline. The inclusion of the discriminator mechanism enables more accurate reasoning and selection, improving performance across different subsets.

Interestingly, SWAP achieves better results on basic algebra compared to more complex topics like Intermediate Algebra or Precalculus, indicating variability in difficulty across different problem types. Meanwhile, the differences in performance between different LLaMA3-8B-Instruct reasoning methods are minor, and direct reasoning appears more effective than few-shot learning for these mathematical problems. Overall, SWAP demonstrates superior mathematical reasoning, particularly

achieving significant improvements in challenging subsets like Intermediate Algebra and Precalculus, highlighting the effectiveness of our approach.

Similarly, in Table 4, SWAP achieved the best performance across all difficulty levels, particularly excelling in the most challenging Level 5, where it reached an accuracy of 14.9%, compared to the best baseline performance of 8.4%. The discriminator mechanism contributes to improved accuracy on high-difficulty problems, demonstrating its effectiveness in enhancing reasoning capabilities. As difficulty increases, all models show a significant decline in performance, particularly at Levels 4 and 5, indicating the increased complexity of reasoning required for these problems. Overall, SWAP consistently outperforms the baseline, especially on higher-difficulty problems, highlighting its advantage in handling complex reasoning tasks.

## F    EFFICIENCY STUDY

In this section, we analyze the efficiency of different planning methods. The time complexity of SWAP is $O(bNT)$, where $b$ is the breadth limit, $N$ is the generation number limit, and $T$ is the step limit. In contrast, the time complexity of RAP (using Monte Carlo Tree Search (MCTS)) (Hao et al., 2023) is $O(N_{\text{sim}}NT)$, where $N_{\text{sim}}$ is the total simulation number limit. Typically, a large number of simulations $N_{\text{sim}} \gg b$ are required to reliably estimate $Q(s, a)$ in MCTS. For ToT (Yao et al., 2023), the time complexity depends on the implementation strategy: 1) Breadth-First Search (BFS): without pruning: $O(N^T)$; with pruning: $O(bNT)$. 2) Depth-First Search (DFS):

Table 5: Efficiency-performance trade-off across different reasoning & planning methods.

|  | GSM8K | |
| --- | --- | --- |
|  | Avg token usage | Acc |
| Llama3-8b-Instruct | | |
| Zero-shot CoT | 175.6 | 70.0 |
| Few-shot CoT | 148.0 | 72.4 |
| ToT | 2214.7 | 75.2 |
| RAP | 5241.4 | 76.0 |
| SWAP (w/o discriminator) | 306.9 | 78.1 |
| SWAP | 3612.0 | 82.7 |

The complexity depends on the state evaluator. The traversal continues until the state evaluator deems the final state satisfactory, making the complexity tied to the evaluation criteria. In conclusion, SWAP is more efficient than RAP and ToT (BFS without pruning version). It is similar to ToT (BFS with pruning version).

We further evaluated the average number of tokens generated using different methods on the GSM8K dataset with the Llama-3-8B-Instruct model. The results are summarized in Table 5. We observed that while the theoretical time complexity of SWAP is comparable to ToT (BFS with pruning), it generates more tokens in practice due to the incorporation of a world model and the construction of an entailment graph. On the other hand, SWAP is significantly more efficient than RAP (MCTS), which involves extensive simulations to reliably estimate the $Q$ value.

## G    EXAMPLES OF SWAP

In this section, we provide example solutions generated by SWAP for all benchmarks used in our paper, including GSM8K, MATH, FOLIO, ReClor, HumanEval, and MBPP. Our framework, SWAP, consists of three main components: the policy model $M_\pi$, the world model $M_{\text{wm}}$, and the controller $M_c$ (Figure 1). Starting with the goal $G$ and the initial state $s_0(\mathcal{G}_0)$, the policy model $M_\pi$ generates an optimized plan $PL$. Using $G$, $PL$, and the current state $s_t(\mathcal{G}_t)$, $M_\pi$ proposes the next action $a_t$ through deliberate planning. After the action is chosen, the world model $M_{\text{wm}}$ predicts the next state $s_{t+1}$ and updates the entailment graph $\mathcal{G}_{t+1}$. Finally, based on $G$ and the updated state $s_{t+1}(\mathcal{G}_{t+1})$, the controller $M_c$ decides whether to continue the reasoning process or output the final answer.

### GSM8K Example

**Problem:** Josh decides to try flipping a house. He buys a house for $80,000 and then puts in $50,000 in repairs. This increased the value of the house by 150%. How much profit did he make?

**Solution:**
"Goal": "Determine how much profit Josh made after flipping the house."

"Initial state": "We know that Josh bought a house for $80,000 and put in $50,000 in repairs, increasing the value by 150%."
"Initial graph": {"Statement": {"s1": "Josh bought a house for $80,000 and put in $50,000 in repairs, increasing the value by 150%."}, "Entailment": {"s1": "Given condition"}}
"Plan": "First, calculate the total investment made by adding the purchase price and repair costs. Then calculate the increased value of the house by multiplying the original value by 1.5. Finally, subtract the total investment from the new value to determine the profit."
"Action 1": "Calculate the total investment by adding $80,000 (purchase price) and $50,000 (repair costs)."
"State 1": "Total investment is $80,000 + $50,000 = $130,000."
"Graph 1": {"Statement": {"s2": "Total investment is $130,000."}, "Entailment": {"s2": ["s1"]}}
"Action 2": "Calculate the increased value of the house by multiplying $80,000 by 1.5."
"State 2": "The increase in value is $80,000 ∗ 1.5 = $120,000."
"Graph 2": {"Statement": {"s3": "The increase in value is $120,000."}, "Entailment": {"s3": ["s1"]}}
"Action 3": "Calculate the new value of the house by adding $80,000 and $120,000."
"State 3": "The new value of the house is $80,000 + $120,000 = $200,000."
"Graph 3": {"Statement": {"s4": "The new value of the house is $200,000."}, "Entailment": {"s4": ["s3"]}}
"Action 4": "Subtract the total investment ($130,000) from the new value of the house ($200,000) to determine the profit."
"State 4": "The profit is $200,000 − $130,000 = $70,000."
"Graph 4": {"Statement": {"s5": "The profit is $70,000."}, "Entailment": {"s5": ["s2", "s4"]}}
"Final answer": "70000"

## MATH Example

**Problem:** Find the distance between the vertex of the graph of the equation $f(x) = x^2 − 8x + 15$ and the point $(0, 2)$.

**Solution:**
"Goal": "Find the distance between the vertex of the parabola and the point $(0, 2)$."
"Initial state": "We need to find the vertex of the parabola defined by the function $f(x) = x^2 − 8x + 15$."
"Initial graph": {"Statement": {"s1": "The function is $f(x) = x^2 − 8x + 15$"}, "Entailment": {"s1": "Given function"}}
"Plan": "To find the vertex of the parabola, we can use the formula for the vertex of a quadratic function in the form $f(x) = ax^2 + bx + c$. Then, we calculate the distance between this vertex and the point $(0, 2)$."
"Action 1": "Find the vertex of the quadratic function $f(x) = x^2 − 8x + 15$."
"State 1": "The vertex of a quadratic function $f(x) = ax^2 + bx + c$ is given by the formula $x = −\frac{b}{2a}$. For $f(x) = x^2 − 8x + 15$:\na = 1, b = −8, and c = 15.\nSubstituting these values:\n$x = −\frac{−8}{2·1} = \frac{8}{2} = 4$\n\nTo find the $y$−coordinate of the vertex, substitute $x = 4$ back into $f(x)$:\n$f(4) = 4^2 − 8 × 4 + 15 = 16 − 32 + 15 = −1$\nThus, the vertex of the parabola is at $(4, −1)$."
"Graph 1": {"Statement": {"s2": "The vertex is $(4, −1)$"}, "Entailment": {"s2": ["s1"]}}
"Action 2": "Calculate the distance between the vertex $(4, −1)$ and the point $(0, 2)$. Use the distance formula."
"State 2": "The distance $d$ between two points $(x_1, y_1)$ and $(x_2, y_2)$ is given by:\n$d = \sqrt{(x_2 − x_1)^2 + (y_2 − y_1)^2}$\nSubstituting $(x_1, y_1) = (4, −1)$ and $(x_2, y_2) = (0, 2)$:\n$d = \sqrt{(0 − 4)^2 + (2 − (−1))^2}$\n$d = \sqrt{(−4)^2 + (2 + 1)^2}$\n$d = \sqrt{16 + 9}$\n$d = \sqrt{25} = 5$\nThe distance is 5."
"Graph 2": {"Statement": {"s3": "The distance between the vertex and the point $(0, 2)$ is 5"}, "Entailment": {"s3": ["s2"]}}
"Final answer": "5"

## FOLIO Example

**Problem:** Determine whether the conclusion is true, false, or uncertain based on the premises provided.

Premises:
When the Monkeypox virus occurs in a being, it may get Monkeypox.
Monkeypox virus can occur in certain animals.
Humans are mammals.
Mammals are animals.
Symptoms of Monkeypox include fever, headache, muscle pains, and tiredness.
People feel tired when they get the flu.

Conclusion: There is an animal.

**Solution:**

"Goal": "Determine whether the conclusion ("There is an animal") logically follows from the given premises."

"Initial state": "Premises: 1. When the Monkeypox virus occurs in a being, it may get Monkeypox. 2. Monkeypox virus can occur in certain animals. 3. Humans are mammals. 4. Mammals are animals. 5. Symptoms of Monkeypox include fever, headache, muscle pains, and tiredness. 6. People feel tired when they get the flu."

"Initial graph": {"Statement": {"Premise 1", "Premise 2", "Premise 3", "Premise 4", "Premise 5", "Premise 6"}, "Entailment": {"Premise 1": "Given condition", "Premise 2": "Given condition", "Premise 3": "Given condition", "Premise 4": "Given condition", "Premise 5": "Given condition", "Premise 6": "Given condition"}}

"Action 1": "Identify any direct connection between the premises and the conclusion."

"State 1": "The conclusion is stating "There is an animal." From Premises 3 and 4, we have that humans are mammals and mammals are animals. Therefore, humans are animals."

"Graph 1": {"Statement": {"s1": "Humans are animals."}, "Entailment": {"s1": ["Premise 3", "Premise 4"]}}

"Action 2": "Verify if any other information from the premises adds to the validity of the conclusion."

"State 2": "Premise 2 states that the Monkeypox virus can occur in certain animals, implying that there are animals susceptible to Monkeypox. This further supports the idea that animals exist."

"Graph 2": {"Statement": {"s2": "Certain animals can get Monkeypox."}, "Entailment": {"s2": ["Premise 2"]}}

"Action 3": "Draw a conclusion based on the gathered information."

"State 3": "From Premises 3, 4, and 2, we can infer that there are animals (humans being a specific type of animal and other animals susceptible to Monkeypox). Therefore, the conclusion "There is an animal" is true."

"Graph 3": {"Statement": {"s3": "There is an animal."}, "Entailment": {"s3": ["s1", "s2"]}}

"Final answer": "True"

---

| ReClor Example |
| --- |

**Problem:** In a business whose owners and employees all belong to one family, the employees can be paid exceptionally low wages. Hence, general operating expenses are much lower than they would be for other business ventures, making profits higher. So a family business is a family's surest road to financial prosperity.

0. ignores the fact that in a family business, paying family members low wages may itself reduce the family's prosperity
1. presumes, without providing justification, that family members are willing to work for low wages in a family business because they believe that doing so promotes the family's prosperity
2. ignores the fact that businesses that achieve high levels of customer satisfaction are often profitable even if they pay high wages
3. presumes, without providing justification, that only businesses with low general operating expenses can succeed

The reasoning in the argument is flawed because the argument

**Solution:**

"Goal": "Identify the flaw in the reasoning presented in the argument."

"Initial state": "We know that the problem presents an argument suggesting that low wages in a family business lead to financial prosperity."

"Initial graph": {"Statement": {"s1": "Family businesses can pay low wages to employees.", "s2": "Lower wages result in lower operating expenses.", "s3": "Lower operating expenses lead to higher profits, implying financial prosperity."}, "Entailment": {"s1": "Given condition", "s2": "Given condition", "s3": "Given condition"}}

"Plan": "We need to evaluate the reasoning used in the argument and determine which option best describes the flaw."

"Action 1": "Analyze the statement about ignoring the impact of low wages on family prosperity (option 0)."

"State 1": "The argument assumes that paying low wages to family members will lead to prosperity without considering the potential negative effect on the family's overall financial well−being. This flaw makes option 0 a plausible conclusion."

"Graph 1": {"Statement": {"s4": "Paying low wages may negatively affect family prosperity."}, "Entailment": {"s4": ["s1"]}}

"Action 2": "Analyze the statement about family members' willingness to work for low wages (option 1)."

"State 2": "The argument assumes that family members are willing to accept low wages to promote prosperity without providing any justification. This makes option 1 another valid flaw."

"Graph 2": {"Statement": {"s5": "Family members are assumed to accept low wages without justification."}, "Entailment": {"s5": ["s1"]}}

"Action 3": "Analyze the statement about high customer satisfaction leading to profitability (option 2)."

"State 3": "The argument does not address whether high levels of customer satisfaction could also lead to profitability, even if wages are high. This makes option 2 a valid flaw as well."
"Graph 3": {"Statement": {"s6": "High customer satisfaction could lead to profitability even with high wages ."}, "Entailment": {"s6": ["s2"]}}
"Action 4": "Analyze the statement about low operating expenses being necessary for success (option 3)."
"State 4": "The argument presumes, without justification, that only businesses with low general operating expenses can be successful. This makes option 3 another potential flaw."
"Graph 4": {"Statement": {"s7": "Only low operating expenses are assumed to lead to success without justification."}, "Entailment": {"s7": ["s3"]}}
"Final answer": "0"

## HumanEval Example

**Problem:**
from typing import List

def has_close_elements(numbers: List[float], threshold: float) −> bool:
    "' Check if in given list of numbers, are any two numbers closer to each other than
    given threshold.
    >>> has_close_elements([1.0, 2.0, 3.0], 0.5)
    False
    >>> has_close_elements([1.0, 2.8, 3.0, 4.0, 5.0, 2.0], 0.3)
    True
    "'

**Solution:**
"Goal": "Write a function `has_close_elements`."
"Initial state": "We have an input list `numbers` of floats and a float `threshold`. We need to check if any two numbers in the list have a difference smaller than `threshold`."
"Initial graph": {"Statement": {"s1": "The input list `numbers` of floats and the float `threshold` are given."}, "Entailment": {"s1": "Given condition"}}
"Plan": "We will sort the list first so that we only need to check adjacent elements. Then, we iterate through the sorted list and check if the absolute difference between any two consecutive numbers is smaller than the `threshold`. If we find such a pair, we return True. If the iteration completes without finding a pair, we return False."
"Action 1": "Sort the input list of numbers."
"State 1": "The list is now sorted, which allows us to only check consecutive elements for small differences."
"Graph 1": {"Statement": {"s2": "The list `numbers` is sorted."}, "Entailment": {"s2": ["s1"]}}
"Action 2": "Iterate through the sorted list and calculate the difference between consecutive elements."
"State 2": "We will stop and return True if we find two consecutive numbers whose difference is smaller than the `threshold`. If no such pair is found, we return False after the iteration completes."
"Graph 2": {"Statement": {"s3": "We check consecutive elements in the sorted list for differences smaller than the threshold."}, "Entailment": {"s3": ["s1", "s2"]}}
"Final answer":
```python
from typing import List

def has_close_elements(numbers: List[float], threshold: float) −> bool:
    # Step 1: Sort the numbers
    numbers.sort()

    # Step 2: Iterate through the sorted list and check differences between consecutive elements
    for i in range(len(numbers) − 1):
        if abs(numbers[i] − numbers[i + 1]) < threshold:
            return True

    # Step 3: Return False if no close elements are found
    return False
```

## MBPP Example

**Problem:** Write a python function to remove first and last occurrence of a given character from the string. Your code should pass these tests:

```
assert remove_Occ("hello", "l") == "heo"
assert remove_Occ("abcda", "a") == "bcd"
assert remove_Occ("PHP", "P") == "H"
```

**Solution:**

"Goal": "Write a function `remove_Occ`."

"Initial state": "We have a string `s` and a character `ch`. We need to find the first and last occurrences of `ch` in `s` and remove them."

"Initial graph": {"Statement": {"s1": "Input string `s` and character `ch` are provided."}, "Entailment": {"s1": "Given condition"}}

"Plan": "We'll first find the index of the first occurrence of `ch` in the string and then the index of the last occurrence. After that, we will remove these two characters and return the modified string."

"Action 1": "Find the indices of the first and last occurrences of `ch` in `s`."

"State 1": "We now have the indices of the first and last occurrences of `ch` in `s`."

"Graph 1": {"Statement": {"s2": "The indices of the first and last occurrences are found."}, "Entailment": {"s2": ["s1"]}}

"Action 2": "Remove the characters at these two indices from the string."

"State 2": "The first and last occurrences of `ch` have been removed from the string."

"Graph 2": {"Statement": {"s3": "First and last occurrences of `ch` are removed."}, "Entailment": {"s3": ["s2"]}}

"Final answer":

```python
def remove_Occ(s: str, ch: str) -> str:
    # Step 1: Find the first occurrence of `ch`
    first_index = s.find(ch)

    # Step 2: Find the last occurrence of `ch`
    last_index = s.rfind(ch)

    # Step 3: Remove the first occurrence of `ch`
    if first_index != -1:
        s = s[:first_index] + s[first_index + 1:]
        # If the first occurrence was before the last, adjust last_index
        if last_index > first_index:
            last_index -= 1

    # Step 4: Remove the last occurrence of `ch`
    if last_index != -1 and last_index != first_index:
        s = s[:last_index] + s[last_index + 1:]

    return s
```

