# OpenReview forum: "Deliberate Reasoning for LLMs as Structure-aware Planning with Accurate World Model"
_ICLR.cc/2025/Conference — Submitted to ICLR 2025_

### Official Review · Reviewer_38qG · 2024-10-25

**Soundness:** 3
**Presentation:** 3
**Contribution:** 3
**Rating:** 6
**Confidence:** 4

**Summary:**

This paper proposes SWAP, which aims to leverage structural information to guide the reasoning process with a world model. The idea is to enable a policy model and a world model to generate diverse actions and states, leading to a higher possibility of getting the correct results. The authors use multiple benchmarks to demonstrate the effectiveness of their proposed method.

**Strengths:**

1. The paper demonstrates that SWAP outperforms various baseline methods on several benchmarks, with ablation studies confirming the effectiveness of the different components proposed by the authors.

2. A key contribution of the paper is framing LLM reasoning as a graph, which allows the model to generate diverse actions and states. This approach helps avoid local minima and enhances verifiability in reasoning, effectively addressing the limitations of Chain-of-Thought (CoT) methods. This contribution is significant for advancing multi-step reasoning.

3. The overall structure of the paper is well-organized; however, there are some weaknesses in presentation that should be addressed.

**Weaknesses:**

1. Many annotations are used before defining them makes the method hard to follow, for example, in line 235 $P_{ori}(z; T_{s_{t-1}})$, it's not clear what is "learned distribution", and the definition of $P_{sem}$ is vague. Annotations are not consistent as well, for example, in line 272 $M_{wm-G}$ is not consistent with $M_{wm}$ in line 175. $M_{\pi_D}$ and $M_{wm-D}$ in equations 8, and 9 are not explained in the paper.
2. One of the contributions is process supervision, however, a comparison to other process supervision methods is missing.
3. The paper proposes to generate diverse actions and states to enhance performance, however, there is a lack of comparison and analysis with diversity-seeking methods using reinforcement learning designed to enhance diversity, such as [1, 2].
4. There lacks a confidence interval for the results, it's not clear how the performance is robust to initialization and randomness.

[1] Vijayakumar, Ashwin K., et al. "Diverse beam search: Decoding diverse solutions from neural sequence models." arXiv preprint arXiv:1610.02424 (2016).

[2] Hu, Edward J., et al. "Amortizing intractable inference in large language models." The Twelfth International Conference on Learning Representations.

**Questions:**

1. How do the semantic similarity is calculated? As mentioned, the method aims to improve the diversity of the reasoning process. However, multi-step reasoning problems can have different orders of actions, leading to semantically different but the same answers. As shown in Figure 1
2.  As shown in Table 1, fine-tuning a Llama-3-8b shows inferior or comparable results to CoT, which is counterintuitive. Could you briefly explain the reason?
3. What is the training cost for SWAP, could the author provide a comparison between SFT and the proposed method?

---

> ### Author Response · Authors · 2024-11-22
> **Rebuttal by Authors to Reviewer 38qG**
>
> Thank you for your insightful comments, which are incredibly helpful in enhancing our work!
>
> ## 1. Notation
> Thanks for your suggestion! We refine the notations throughout the framework, and add the detailed pseudo-code (Algorithm 1 and 2). Please refer to the revised version (Section 4.1, 4.2, and 4.3) for more details.
>
> ## 2. Comparison to process reward model
> We add the following content in the related work section.
>
> Although recent research has increasingly explored automatic process annotations using tree search, training an effective PRM remains challenging, as from a mathematical perspective, it assigns a **numerical value** within $[0, 1]$ to each state **independently**.
> To overcome this problem, we propose a novel strategy for **automatic ranking annotation**, i.e., given the current context and a set of candidate options, selecting the best option based on relative quality.
> Our ranking strategy offers significant advantages over traditional PRMs: 1) it emphasizes relative quality, making it more robust to noise; 2) it simplifies optimization and enhances generalization.
> Notably, our high-quality automatic ranking annotation method is non-trivial as it systemically incorporates three key factors: 1) **structural information**; 2) **correctness**; and 3) **semantical equivalence**.
>
> We also add experiments with PRMs (PRM800k [1] and Math-Shepherd [2]) with the same baseline model in the revised version (Table 1). It shows that the performance of our method is better than those of PRMs.
>
> ## 3. Comparison to other diversity-seeking methods
> Compared to related work [3,4], our approach offers the following advantages:
>
> 1) **Diverse beam search** [3] performs beam search in groups using a diversity-augmented objective. However, it has several limitations: 1) Beam search at the token level is computationally intractable; 2) Searching the most suitable strategies and hyperparameters for similarity calculation can be time-consuming; 3) For reasoning tasks involving special tokens (e.g., math reasoning or first-order logic reasoning), embedding-based similarity calculations may be unreliable.
>
>    In contrast, **SWAP** (implemented as SFT with LoRA) employs an end-to-end learning paradigm, leveraging the world knowledge in pre-trained LMs. Our generation process is sampling-based, making it more efficient than token-level beam search.
>
> 2) **GFlowNets fine-tuning** [4] is a diversity-seeking reinforcement learning algorithm that uses amortized Bayesian inference. Although it demonstrates better performance compared to SFT with limited training data, it is unclear whether it can scale to large-scale  datasets and complex reasoning tasks. As a reinforcement learning method, GFlowNets fine-tuning can be significantly more challenging and costly to train when dealing with large-scale datasets.
>
>     In contrast, **SWAP** is more scalable and better suited for handling large-scale datasets and complex reasoning tasks efficiently.
>
> We have summarized the above advantages and added to the revised version (Section 4.2).
>
> ## 4. Confidence interval
> We have added confidence interval for the results in Table 1. Please refer to the revised version.
>
> ## 5. Calculation of semantic similarity
> We have added the calculation process in the revised version (Section 4.2).
>
> The effect of action order in multi-step reasoning is an interesting question. Currently, our approach focuses on increasing diversity step-by-step. In our experiments, we observed that the order of actions typically remains stable, since the actions are often interdependent.
>
> As future work, this issue could be addressed through post-processing filtering based on similarity calculations.
> Specifically, if a generated trajectory is too similar to any existing trajectories in the stored pool, it will be discarded.
> To compare trajectories, the proposed graph representation provides a significant advantage, as graph similarity algorithms can be effectively utilized for this task.
> Additionally, we can explore data augmentation by reordering the actions based on the graph's dependency structure, further enhancing diversity and robustness.

---

> ### Author Response · Authors · 2024-11-23
> **Continuation of Rebuttal by Authors to Reviewer 38qG**
>
> ## 6. Performance of SFT on CoTs
> The performance of SFT on CoTs slightly decreases for certain datasets (MATH, HumanEval, MBPP). For these datasets, we fine-tune the model using the provided reasoning processes (or code) written by humans. Language models typically benefit from training data with more steps (tokens) that detail the thinking process. However, we observed that human-written answers are often more concise, omitting some intermediate steps and details.
>
> This results in a reduction in the number of reasoning tokens (during inference) after fine-tuning, which negatively impacts the model's performance. As noted by OpenAI, there exists an inference-time scaling law: models achieve higher accuracy with more inference tokens.
>
> Additionally, it is important to note that these pre-trained models have likely already been fine-tuned on these datasets. Re-fine-tuning on the same datasets yields marginal benefits and can even degrade performance.
>
>
> ## 7. Training cost
> As discussed in Section 4.2 and illustrated in Figure 2 (the revised version), our approach is implemented using SFT with LoRAs, with the additional cost arising from training a semantic equivalent LoRA. Specifically, starting from the original steps, we generate semantically equivalent alternatives by prompting GPT-4o. The semantic equivalent LoRA is then fine-tuned using the collected training data. For each trajectory, we sample some steps and generate two alternatives for each step. Consequently, the additional training cost of our method is no more than twice that of SFT with LoRA on the trajectory, resulting in a total cost of less than three times the standard SFT with LoRA.
>
>
> ## Reference:
> [1] Lightman, Hunter, et al. "Let's verify step by step." arXiv preprint arXiv:2305.20050 (2023).
>
> [2] Wang, Peiyi, et al. "Math-shepherd: A label-free step-by-step verifier for llms in mathematical reasoning." arXiv preprint arXiv:2312.08935 (2023).
>
> [3] Vijayakumar, Ashwin K., et al. "Diverse beam search: Decoding diverse solutions from neural sequence models." arXiv preprint arXiv:1610.02424 (2016).
>
> [4] Hu, Edward J., et al. "Amortizing intractable inference in large language models." The Twelfth International Conference on Learning Representations.

---

> > ### Comment · Reviewer_38qG · 2024-11-25
> > **Thanks for the response from authors**
> >
> > The revision makes it more clear than the original version, and thanks to the answer from the authors, I would like to increase the rating.

---

> > > ### Author Response · Authors · 2024-11-25
> > > **Thanks for reply**
> > >
> > > Thank you for your acknowledgment and support!

---

### Official Review · Reviewer_zS5k · 2024-11-03

**Soundness:** 2
**Presentation:** 1
**Contribution:** 2
**Rating:** 5
**Confidence:** 2

**Summary:**

The paper discusses a framework called Structure-aware Planning with Accurate World Model (SWAP) that aims to enhance the reasoning capabilities of large language models (LLMs). Key designs include:

**Framework Overview**: SWAP integrates structural information into the reasoning process, providing a soft verification mechanism that guides LLMs through multi-step reasoning. The authors suggest this approach may improve upon traditional Chain-of-Thought (CoT) reasoning methods, which can lack effective verification mechanisms.

**Generator-Discriminator Architecture**: The framework employs a Generator-Discriminator architecture to enhance world modeling. The generator is responsible for predicting future states, while the discriminator evaluates these predictions to improve the accuracy of the reasoning process.

**Diversity-Based Modeling**: The paper introduces a method to encourage diversity in action generation and state prediction, which is intended to allow the model to explore a broader range of solutions and avoid premature convergence on suboptimal paths.

**Contrastive Ranking for Discrimination Accuracy**: The authors implement a contrastive ranking approach that focuses on relative comparisons to improve the discriminator's ability to identify valid and invalid reasoning steps.

**Strengths:**

- The introduction of normalization metrics (Eq. 2 & 4) for Diversity-Based Modeling is novel and interesting.
- The method is shown effective on diverse tasks (math, coding and logical reasoning) with Llama3-8b as backbone.

**Weaknesses:**

- The paper writing lacks clarity. Especially about the structured search algorithm as mentioned in 4.1. The generator and discriminator framework only shows how to select the action when more than one choices are provided. However, when more than one state and action are selected, what is the next state to process among multiple parallel choices? Figure 2 illustrates the process as a linear step by step process and fails to present the tree search as shown in Figure 4. It would be better if Figure 2 is replaced by detailed pseudo code.

- The framework of this algorithm is very similar to related work (TOT, RAP and [1]). RAP also uses world model based tree search and the main difference is MCTS style search is used to rank choices while this work uses a discriminator to reject choices. [1] proposes a similar pipeline that also uses discriminator-aided tree search. Wonder if the authors could point out the main difference with these works and provide a elaborated compare of the main pipeline.

- Figure 3 plots the use of LORA tuning to improve generation diversity. However, only posthoc adjustment methods are proposed in the section. Wonder if the authors could further explain this figure.

- Table 1 lists a lot of redundant backbone models that may not be directly comparable with SWAP + Llama3-8b. It would be better if the authors compare algorithms while fixing backbones and add more comparison with different backbone models.

[1]Chen, Ziru, et al. "When is tree search useful for llm planning? it depends on the discriminator." arXiv preprint arXiv:2402.10890 (2024).

**Questions:**

1. What is the efficiency of the SWAP tree search algorithm compared to TOT and RAP? SWAP uses discriminator to prune many options and may have an advantage in efficiency.

---

> ### Author Response · Authors · 2024-11-22
> **Rebuttal by Authors to Reviewer zS5k**
>
> Thank you for your insightful and constructive comments, which are greatly appreciated and extremely helpful in enhancing our work!
>
> ## 1. Clarity
> Thanks for your suggestion! Figure 2 is mainly used to illustrate the input and output of different modules. We replace it with the detailed **pseudo-code** in the revised version (Algorithm 1 and 2). Additionally, we modify **Figure 1** to depict the step-by-step process of graph construction. To enhance clarity, we also refine the **notations** throughout the framework. Please refer to the revised version (Section 4.1) for the updated details.
>
> ## 2. Difference from related work
> The key innovation in our framework is viewing multi-step reasoning as the process of **entailment graph construction**, supported by a **structure-aware planning** approach tailored for this purpose.
>
> Using graph representation offers several advantages: (1) It explicitly captures the **non-linear structure** of the reasoning process, providing LMs with enhanced guidance; (2) It allows for greater **control** over the reasoning process. For instance, graph representation facilitates structural verification during training data collection and enhances discrimination accuracy during inference. (3) It also enables exciting future work, such as exploring **the impact of action order** in multi-step reasoning. By capturing the dependency between steps, it becomes possible to identify interchangeable steps for data augmentation. Furthermore, for **long-context reasoning** tasks (e.g., OpenAI's o1 model), a graph structure can provide improved understanding and better control over the process.
>
> Given the representation, we further notice that existing methods (ToT, RAP) mainly uses **prompt engineering** which totally or partially rely on **self-evaluation**. These strategies bring limited benefits in complex reasoning tasks since self-evaluation without external feedback can be unreliable [2,3]. To overcome this challenge, we propose using a generator-discriminator structure with fine-tuning, and further identify the bottlenecks of generation diversity and discrimination accuracy. We address these bottlenecks with **architecture-level adaptation**.
> This generator-discriminator structure also contributes to obtaining an **accurate** world model which is not considered in related work (RAP).
>
> As for [1], it mainly focuses on **code generation** (text-to-SQL Parsing, Python-program-based math reasoning), and relies on **external feedback** (i.e., program execution results). Unlike [1], we do not constrain the model on code generation which could hurt its expressiveness on complex reasoning tasks [4]. It also does not consider architecture-level adaptation for generation diversity and discrimination accuracy.
>
> All these strategies distinguish our framework from related work and contribute to **substantial improvements** (see TOT, RAP in Table 1).
>
> ## 3. Explanation of generation diversity calculation
>
> We have included the detailed calculation process in the revised version (Section 4.2). Please refer to it for further details.
>
> ## 4. Comparison to existing methods with different base models
> Thanks for your suggestion! We have delete the methods in Table 1 that uses different base models. We now test different methods with the **same** base model (Llama 3-8b-instruct). We also add **PRM** methods (PRM800k [5] and Math-Shepherd [6]). In addition, we change the base model into **Mistral-7b-instruct** and test all the methods. The results (Table 1) show that our method consistently outperform existing methods.
>
> ## 5. Efficiency
>
> The time complexity of **SWAP** is $O(bNT)$, where $b$ is the breadth limit, $N$ is the generation number limit, and $T$ is the step limit. In contrast, the time complexity of **RAP** (using MCTS) is $O(N_{sim}NT)$, where $N_{{sim}}$ is the total simulation number limit. Typically, a large number of simulations $N_{{sim}} \gg b$ are required to reliably estimate $Q(s, a)$.
> For **ToT**, the time complexity depends on the implementation strategy: 1) Breadth-First Search (BFS): without pruning: $O(N^T)$; with pruning: $O(bNT)$. 2) Depth-First Search (DFS): The complexity depends on the state evaluator. The traversal continues until the state evaluator deems the final state satisfactory, making the complexity tied to the evaluation criteria.
>
> In conclusion, SWAP is more efficient than RAP and ToT (BFS without pruning version). It is similar to ToT (BFS with pruning version).

---

> > ### Author Response · Authors · 2024-11-22
> > **Continuation of Rebuttal by Authors to Reviewer zS5k**
> >
> > ## Reference:
> > [1] Chen, Ziru, et al. "When is tree search useful for llm planning? it depends on the discriminator." arXiv preprint arXiv:2402.10890 (2024).
> >
> > [2] Huang, Jie, et al. "Large language models cannot self-correct reasoning yet." arXiv preprint arXiv:2310.01798 (2023).
> >
> > [3] Jiang, Dongwei, et al. "Self-[in] correct: Llms struggle with refining self-generated responses." arXiv preprint arXiv:2404.04298 (2024).
> >
> > [4] Yang, Yuan, et al. "Can LLMs Reason in the Wild with Programs?." arXiv preprint arXiv:2406.13764 (2024).
> >
> > [5] Lightman, Hunter, et al. "Let's verify step by step." arXiv preprint arXiv:2305.20050 (2023).
> >
> > [6] Wang, Peiyi, et al. "Math-shepherd: A label-free step-by-step verifier for llms in mathematical reasoning." arXiv preprint arXiv:2312.08935 (2023).

---

> ### Comment · Reviewer_zS5k · 2024-11-24
> **Thank you for your response**
>
> Thanks to the authors for their efforts to improve the paper. The addition of pseudocode and the revised Table 1 make the paper much clearer than the original draft. Therefore, I would like to raise my rating to borderline.
>
> I still have two remaining questions:
>
> - Can these trained scores be used as reward models, similar to Math-Shepherd?
> - The efficiency comparison may not be entirely accurate when based solely on bounds, as there could be differences in coefficients. Could the authors provide an estimate of the real-time ratio for different methods? For example, this could be based on the average number of tokens generated per test case.

---

> > ### Author Response · Authors · 2024-11-25
> > **Thanks for reply**
> >
> > Thanks for your reply and recognition!
> >
> > As for the remaining questions:
> >
> > ### 1. Reward model
> > In a Markov Decision Process (MDP), a reward model (or score function) is crucial for learning a policy.
> > Although recent research (Math-Shepherd) has increasingly explored automatic process annotations, training an effective PRM remains challenging, as from a mathematical perspective, it assigns a numerical value within $[0, 1]$ to each state independently.
> >
> > To overcome this problem, we propose a novel strategy for automatic ranking annotation, i.e., given the current context and a set of candidate options, selecting the best option based on relative quality.
> > Our ranking strategy offers significant advantages over traditional PRMs: 1) it emphasizes relative quality, making it more robust to noise; 2) it simplifies optimization and enhances generalization.
> > Notably, our high-quality automatic ranking annotation method is non-trivial as it systemically incorporates three key factors: 1) structural information; 2) correctness; and 3) semantical equivalence.
> >
> > We also add experiments with PRMs (PRM800k and Math-Shepherd) with the same baseline model (Table 1). It shows that the performance of our method is better than those of PRMs.
> >
> > ### 2. Efficiency comparison
> > We evaluated the average number of tokens generated using different methods on the GSM8K dataset with the Llama-3-8B-Instruct model. The results are summarized as follows:
> >
> > | Method | Avg token usage | Accuracy |
> > |----------|----------|----------|
> > | Zero-shot CoT   | 175.6   | 70.0   |
> > | ToT  | 2214.7  | 75.2  |
> > | RAP    | 5241.4   | 76.0   |
> > | SWAP (w/o discriminator)    | 306.9   | 78.1   |
> > | SWAP    | 3612.0   | 82.7 |
> >
> > We observed that while the theoretical time complexity of SWAP is comparable to ToT (BFS with pruning), it generates more tokens in practice due to the incorporation of a world model and the construction of an entailment graph. On the other hand, SWAP is significantly more efficient than RAP (MCTS), which involves extensive simulations to estimate the $Q$-value.

---

> > > ### Comment · Reviewer_zS5k · 2024-11-25
> > > **Thank you**
> > >
> > > Thanks to the the authors for the clarifications. The efficiency/performance trade-off of SWAP appears favorable. It would be great to include this analysis in the new version.

---

> > > > ### Author Response · Authors · 2024-11-25
> > > > **Thank you for your suggestion**
> > > >
> > > > Thank you for your suggestion!
> > > >
> > > > We have added this section in the revised version (Appendix F) to further highlight the advantages of our method compared to related work.

---

> > > > > ### Author Response · Authors · 2024-11-27
> > > > > **Look forward to address remaining concerns**
> > > > >
> > > > > We have made modifications based on your suggestions and would like to know if you have any remaining concerns. We are eager to address them promptly!

---

> > > > > > ### Author Response · Authors · 2024-12-01
> > > > > > **Follow-Up on Revisions for Reviewer zS5k**
> > > > > >
> > > > > > Thank you for your valuable feedback. We have addressed your suggestions in the revised version.
> > > > > >
> > > > > > We hope these changes address your concerns and demonstrate the improvements. Please let us know if you have any further comments or suggestions.

---

### Official Review · Reviewer_F56z · 2024-11-04

**Soundness:** 2
**Presentation:** 3
**Contribution:** 3
**Rating:** 5
**Confidence:** 3

**Summary:**

This paper proposes a framework for improving reasoning in LLMs. The framework consists of a graph representation of the multi-step reasoning of the problem, a generator to generate possible next steps, and a discriminator to rank the possible solutions generated by the generator. The paper proposes several improvements to the framework such as adding diversity to step generation, improving the discriminator via process-supervision, and adding meta-knowledge of the problem into the LLM. The paper then shows the effectiveness of the framework by showing improved results on various mathematical reasoning and coding benchmarks.

**Strengths:**

* The paper is well-written and conveys the overall framework well.
* Representing step-by-step reasoning as a graph is an interesting idea.
* The ablation study shows the performance gain of each design choice.

**Weaknesses:**

* The related work section is very thin. The paper should more thoroughly compare its framework with existing reward modeling and process supervision literature. For instance, how does this method (and its performance) compare with [1]? Especially since the authors take inspiration from such works, they should also include some process-supervision methods in their Llama3-8B baselines.

* The reported benchmark numbers for Llama3-8B are significantly lower than what the official llama has reported (e.g., according to Llama's report, HumanEval zero-shot should be 62.2, and MATH 4-shot-CoT should be 30.0, and GSM8K 8-shot-CoT is 79.6). This casts doubt on the validity of the evaluation done by the authors, particularly for HumanEval, where the original Llama's reported performance is higher than what SWAP achieves.

* There is very little discussion on the advantage of having a graph (with node connectivities) rather than sequentially listing all the reasoning steps. Can the authors provide any ablations on this? How well is the LLM able to parse the dependencies through json node connectivity? Can the authors provide an example of how the LLM reasoning for a few problems, and what the final graph produced by the LLM would look like?


[1] Wang et al, Math-Shepherd: Verify and Reinforce LLMs Step-by-step without Human Annotations, 2023

**Questions:**

* Current results consider Llama3-8B generalist model. Is the proposed method able to improve math-specialized (e.g., Qwen Math) or code-specialized (e.g., Deepseek Coder) models?

---

> ### Author Response · Authors · 2024-11-22
> **Rebuttal by Authors to Reviewer F56z**
>
> Thank you for your insightful comments; they are extremely valuable for enhancing our work!
>
> ## 1. Related work
> We add the following context in the revised version (related work section) to compare our framework with existing reward modeling and process supervision literature.
>
> There are two primary types of reward models: Outcome Reward Model (ORM) and Process Reward Model (PRM).
> The ORM evaluates the fully generated solution by assigning a single scalar confidence score.
> Its training relies on outcome supervision by comparing generated answers with the ground truth.
> In contrast, the PRM provides stepwise rewards throughout the reasoning process, assigning a scalar confidence score to each intermediate steps.
> Empirical evidence shows that, compared with outcome supervision, process supervision ensures the correctness of each step, providing more benefits to multi-step reasoning.
> However, the training of PRM requires process supervision, which is hard to obtain, e.g., collecting process annotation from humans is inherently not scalable.
> Although recent research has increasingly explored automatic process annotations using tree search, training an effective PRM remains challenging, as from a mathematical perspective, it assigns a **numerical value** within $[0, 1]$ to each state **independently**.
> To overcome this problem, we propose a novel strategy for **automatic ranking annotation**, i.e., given the current context and a set of candidate options, selecting the best option based on relative quality.
> Our ranking strategy offers significant advantages over traditional PRMs: 1) it emphasizes relative quality, making it more robust to noise; 2) it simplifies optimization and enhances generalization.
> Notably, our high-quality automatic ranking annotation method is non-trivial as it systemically incorporates three key factors: 1) **structural information**; 2) **correctness**; and 3) **semantical equivalence**.
>
> We also add experiments with PRMs (PRM800k [1] and Math-Shepherd [2]) with the same baseline model in the revised version (Table 1). It shows that the performance of our method is better than those of PRMs.
>
> ## 2. Performance gap
> We checked the evaluation details of the LLama3 model [3,4,5]. The reported results actually come from **majority voting** rather than single-time inference. Specifically, for the same question, they generate 100 solutions for GSM8k, 256 solutions for MATH, and 200 solutions for HumanEval. In addition, it is possible that they perform some **hyper-parameter optimization** (e.g., seed, temperature, topK, topP). We also found that since the **test set size** of HumanEval is relatively small (164), the large performance gap is actually responding to a few test samples.
>
> It would be too expensive to replicate the same results. Thus, for practical considerations, we use the default evaluation setting (single-time inference, low temperature=0.2, fixed seed) in our experiments for all methods to ensure a fair comparison.
>
> ## 3. Advantages of Graph Representation
> We add the following content in the revised version (related work section).
>
> Furthermore, we notice that although some reasoning processes are inherently **non-linear**, existing methods mainly follow a linear problem-solving manner.
> Language models are expected to implicitly infer the non-linear structure from the linear representation of the reasoning process, which proves challenging for complex reasoning tasks.
> To help the model, we integrate **structural information** into the reasoning process which explicitly represents the reasoning structure within the context.
> These structures provide the language model with additional **guidance** and **control**, enabling extra capabilities such as symbolic learning and verification.
>
> We emphasize that our approach replaces the original state with a state-graph pair, consisting of a natural language description and its corresponding graph representation. The graph structure serves as an explicit representation of the non-linear dependency between steps. The effectiveness of integrating graph representations is already demonstrated through an ablation study (Table 2 'w/o structure info').
>
> For parsing, we explored various formats and finalized using a JSON dictionary, where child indices serve as keys and lists of parent indices as values. During training data collection, we provided demonstrations in this format to GPT-4o and applied ad-hoc filtering to ensure high data quality. After fine-tuning, the base model demonstrated the ability to effectively generate and interpret this format. The good performance of our method (with ablation study) also demenstrate the parsing capability of LMs.
>
> We already provide multiple example outputs in the paper (Appendix F).

---

> ### Author Response · Authors · 2024-11-22
> **Continuation of Rebuttal by Authors to Reviewer F56z**
>
> ## 4. Our approach on specialized models
>
> This is an interesting question! Given the time constraints, we primarily conducted experiments using the DeepSeek-Math-7B-Instruct model and observed that SWAP also performs effectively with this specialized model. Expanding the evaluation to include additional specialized models is planned as part of future work.
>
> |  | GSM8K | MATH |
> |----------|----------|----------|
> | **Deepseek-math-7b-Instruct** |
> | CoT    | 82.0   | 45.4   |
> | SWAP (w/o discriminator)    |  82.4  | 45.0   |
> | SWAP  | 86.1 | 47.5   |
>
> ## Reference:
> [1] Lightman, Hunter, et al. "Let's verify step by step." arXiv preprint arXiv:2305.20050 (2023).
>
> [2] Wang, Peiyi, et al. "Math-shepherd: A label-free step-by-step verifier for llms in mathematical reasoning." arXiv preprint arXiv:2312.08935 (2023).
>
> [3] https://github.com/meta-llama/llama3/blob/main/eval_details.md
>
> [4] Dubey, Abhimanyu, et al. "The llama 3 herd of models." arXiv preprint arXiv:2407.21783 (2024).
>
> [5] Touvron, Hugo, et al. "Llama: Open and efficient foundation language models." arXiv preprint arXiv:2302.13971 (2023).

---

> > ### Comment · Reviewer_F56z · 2024-11-24
> >
> > Thank you for the newly added comparisons.
> >
> > > We also add experiments with PRMs (PRM800k [1] and Math-Shepherd [2]) with the same baseline model in the revised version (Table 1). It shows that the performance of our method is better than those of PRMs.
> >
> > What is the exact details of training with the PRM800K dataset? from my understanding it contains significant fraction of the MATH test set, except for only 500 samples (MATH500), how is the evaluation comparison done on the entire MATH test set?
> >
> > > We checked the evaluation details of the LLama3 model [3,4,5]. The reported results actually come from majority voting rather than single-time inference. Specifically, for the same question, they generate 100 solutions for GSM8k, 256 solutions for MATH, and 200 solutions for HumanEval.
> >
> > The link in [3] (provided by the authors) explicitly mentions that for MATH and GSM8K the results are maj@1, can you point out to the reference of maj@100 and maj@256 you are referring to?
> >
> > Also, what does the CoT-SC in Table 1 refer to? From my understanding, it should be the self-consistency/majority sampling baseline (with how many samples?).
> >
> > > we use the default evaluation setting (single-time inference, low temperature=0.2, fixed seed) in our experiments for all methods to ensure a fair comparison.
> >
> > The default evaluation setting for maj@1 is typically greedy sampling (i.e.,  temperature=0), which may be one reason behind the author’s low numbers.
> >
> > > We also found that since the test set size of HumanEval is relatively small (164), the large performance gap is actually responding to a few test samples.
> >
> > This does not justify the performance gap, or the 12% lower performance compared to the report of the original llama.

---

> ### Author Response · Authors · 2024-11-24
> **Rebuttal by Authors to Reviewer F56z**
>
> Thank you for the follow-up questions.
>
> ## 1. Training on PRM800k
> To maintain consistency with the original experiments (which were conducted on the complete test set of MATH), we also evaluate PRM (trained on PRM800k) on the full test set.
> We observed that the original PRM800k data [1] includes 4,500 test samples from the MATH dataset in its training data.
> To address this issue, we follow the approach in [2] and use their code [3] to fine-tune Llama3-8b-instruct on a filtered PRM800k dataset as a PRM.
> Specifically, we remove samples from the training set that belong to the test split of MATH.
>
> ## 2. Evaluation details of official Llama3
> From our understanding, [4] uses the vague term "maj@1" to describe the evaluation process for GSM8k and MATH.
> If only one-time inference is conducted, the term "maj" would be unnecessary (as seen with other benchmarks in [4]).
> Furthermore, we observed that Llama3 largely follows the evaluation settings of Llama1 for many benchmarks (as mentioned in [4]).
> To verify, we referred to the Llama1 paper [5] and found that it employs majority voting with 256 samples for MATH and 100 samples for GSM8k (Table 7 in [5]).
> Based on our experimental results, we believe Llama3 uses the same majority voting settings.
> Also, we only use 4-shot CoT for all benchmarks (including GSM8k), in contrast to the 8-shot CoT utilized in the official report.
>
> The CoT-SC in Table 1 refers to Chain-of-Thought reasoning with majority voting based on 8 samples.
>
> ## 3. Effect of temperature
> We set the temperature to 0.2 as planning methods require diversity. We want to maintain this setting for a fair comparison.
>
> Additionally, we conducted experiments on HumanEval using different temperatures, with the results as follows:
>
> | Llama3-8b-Instruct (0-shot)  | 1  | 2  | 3  | 4  | 5  | 6  | 7  | 8  | 9 | 10 | avg. |
> |--------|--------|--------|--------|--------|--------|--------|--------|--------|--------|--------|--------|
> | temperature = 0  |   52.4|53.0|51.8|51.8|52.4|51.8|51.8|52.4|53.0|53.0|52.4|
> | temperature = 0.2  | 48.2  | 51.2|51.8|49.4|47.6|52.4|49.4|51.2|49.4|51.2|50.2|
>
> We observed that a lower temperature (0) leads to more stable results. However, the average accuracy remains at 52.4, which is close to 50.2 (reported in the paper). We believe that demonstrating the superiority of the proposed method under the same fair comparison setting is sufficient.
>
> ## 4. Performance gap on HumanEval
> We provide the generated results on HumanEval from Llama3-8b-instruct (0-shot) for further verification.
>
> Anonymous cloud storage:
> https://drive.google.com/drive/folders/1nSg15osa-dwBTYPuepyYT1NiyCXOSgie?usp=sharing
>
>
> ## Reference
>
> [1] https://github.com/openai/prm800k/tree/main
>
> [2] Sun, Zhiqing, et al. "Easy-to-hard generalization: Scalable alignment beyond human supervision." arXiv preprint arXiv:2403.09472 (2024).
>
> [3] https://github.com/Edward-Sun/easy-to-hard
>
> [4] https://github.com/meta-llama/llama3/blob/main/eval_details.md
>
> [5] Touvron, Hugo, et al. "Llama: Open and efficient foundation language models." arXiv preprint arXiv:2302.13971 (2023).

---

> ### Author Response · Authors · 2024-11-25
> **Rebuttal by Authors to Reviewer F56z**
>
> To avoid confusing the audience, we have added clarifications in Table 1 to better convey the implementation details. Specifically, we have updated method names as follows: "Few-shot CoT" has been changed to "Few-shot CoT (4-shot)", "CoT-SC" has been updated to "SC@maj8", and "PRM (PRM800K)" is now labeled as "PRM (PRM800K*)". Additionally, we have clarified in the table caption with the note: *"We use the filtered PRM800K dataset [1] to evaluate performance on the full MATH test set."* Please refer to the revised version.
>
> Another potential issue is the comparison between low-temperature sampling (e.g., 0.2) and greedy decoding. Given the time constraints, our experiments were primarily conducted on HumanEval. We observed that while greedy decoding produces more stable results, the performance differences between the two methods are minimal. In the next version of this work, we plan to conduct experiments across all benchmarks and include this analysis in the appendix.
>
> We hope these modifications effectively address your concerns!
>
> [1] Sun, Zhiqing, et al. "Easy-to-hard generalization: Scalable alignment beyond human supervision." arXiv preprint arXiv:2403.09472 (2024).

---

> > ### Author Response · Authors · 2024-11-27
> > **Rebuttal by Authors to Reviewer F56z**
> >
> > We further investigated the performance gap between our experimental results and the official LLaMA report.
> >
> > Upon analyzing the model outputs, we observed that with the few-shot CoT method, the model sometimes repeats example code snippets provided in the prompt, leading to errors during evaluation.
> > To address this issue, we conducted verification experiments to refine and identify the most suitable instructions for few-shot CoT. Similarly, we reviewed and optimized the instructions used for zero-shot CoT.
> >
> > Considering the potential impact of temperature and instructions, we re-evaluated the zero-shot CoT, few-shot CoT, and SFT-CoT methods on the HumanEval and MBPP benchmarks for all base models.
> > This evaluation utilized greedy decoding and optimized instructions.
> > The updated results show significant improvements, with the 4-shot CoT achieving a score of 0.568 on HumanEval, which is much closer to the official LLaMA result of 0.622.
> > Please refer to the revised version (Table 1) for the detailed results.
> >
> > We hope these modifications effectively address your concerns!

---

> > > ### Comment · Reviewer_F56z · 2024-11-27
> > >
> > > Thank you for your efforts in closing the gap between your reported numbers and the original works. I am glad that the rebuttal discussion has helped strengthen the considered baselines.
> > >
> > > > Specifically, we remove samples from the training set that belong to the test split of MATH.
> > >
> > > I would suggest the authors report the remaining dataset size. Also, I could not find the data sizes the authors generate (from GPT-4 and DeepSeek) for training their own method, which I suggest they add.
> > >
> > > Also, I suggest the authors mention how many roll-outs/samples (i.e., RM@K) they take from both SWAP and PRM methods in Table 1.
> > >
> > > > From our understanding, [4] uses the vague term "maj@1" to describe the evaluation process for GSM8k and MATH
> > >
> > > maj@1 has a widely understood meaning in the community.
> > >
> > > Overall, I find the experimental results that, a LoRA-tuned SWAP is outperforming the best PRM model by 13.5% on the MATH dataset, to be very strong. Such strong results require rigorous and strong evaluation, and given the rebuttal discussion, it seems that the baselines lack very basic tuning (such as temperature and maj@k), which casts double on the whole evaluation pipeline. I tried to take a look at the code, but it is very unreadable and lacks a README file.
> > >
> > > Therefore, I maintain my score.

---

> > > > ### Author Response · Authors · 2024-12-01
> > > > **Rebuttal by Authors to Reviewer F56z**
> > > >
> > > > Thank you for your suggestions and concerns. We address them in detail below:
> > > >
> > > > ## Implementation details
> > > >
> > > > The filtered PRM800k training set contains 521,007 samples. For a fair comparison with SWAP, we randomly selected 100,000 annotations, matching the scale of SWAP.
> > > >
> > > > For SWAP:
> > > > the number of trajectories for the generator are as follows: GSM8k (28.3k), MATH (49.3k), ReClor (14.5k), FOLIO (7.3k), HumanEval (3.1k), and MBPP (1.3k);
> > > > the number of semantically equivalent pairs we obtained are as follows: GSM8k (8.1k), MATH (24.2k), ReClor (7.1k), FOLIO (3.8k), HumanEval (1.6k) and MBPP (0.7k);
> > > > the number of process annotations for the discriminator are as follows: GSM8k (48.0k), MATH (98.2k), ReClor (28.7k), FOLIO (14.1k), HumanEval (6.0k), and MBPP (2.5k).
> > > >
> > > > The number of candidate solutions for PRMs is set to 8.
> > > > For SWAP, the number of rollouts (breadth limit) is set as 8.
> > > >
> > > > We have incorporated these details into the paper. Please refer to the revised version in Section 5.1 and Appendix D.
> > > >
> > > > ---
> > > > Given that our setting differs from the conventional PRM setup, we provide the following justifications:
> > > >
> > > > - **Training set filtering**: The original PRM800k dataset uses 90% of the MATH test split. If we do not filter their dataset, we need to create extra training data for SWAP with the same 90% test split to ensure a fair comparison.
> > > >
> > > > - **Training set sampling**: The process annotation density of the filtered PRM800k (521k) is significantly higher than that of SWAP (98k). Without further sampling, a fair comparison cannot be achieved.
> > > >
> > > > - **Practical consideration**: This experiment is an addition. Training with the original dataset is infeasible due to time constraints.
> > > >
> > > > ## Reproducing official Llama3 performance
> > > >
> > > > We wish to draw the reviewer's attention to the following points:
> > > >
> > > > 1) The official Llama release does not include evaluation code. [1] provides only a brief description (one or two sentences) of the evaluation process for each benchmark.
> > > >
> > > > 2) The official Llama results involve hyperparameter optimization tailored to specific benchmarks. For example, they report results [2] for Llama-3-8B-Instruct on MMLU (5-shot), GPQA (0-shot), HumanEval (0-shot), GSM-8K (8-shot, CoT), and MATH (4-shot, CoT) with different settings.
> > > >
> > > > Considering 1 and 2, reproducing the official results is a non-trivial task and falls beyond the scope of our research. Our objective is to approximate the official results as closely as possible within our available resources.
> > > >
> > > > ---
> > > > **Current results:**
> > > >
> > > > GSM8K: ours 73.7 (4-shot, CoT, greedy) v.s. official 79.6 (8-shot, CoT, maj@1)
> > > >
> > > > MATH: ours 28.2 (0-shot, CoT, greedy) v.s. official 30 (4-shot, CoT, maj@1)
> > > >
> > > > ## Baseline fine-tuning
> > > >
> > > > **1. Temperature (T):**
> > > > Temperature may influence performance to some extent but does not affect the overall conclusion.
> > > > We have already presented results on HumanEval, where 0-shot performance improved from 50.2 (T=0.2) to 52.4 (T=0).
> > > > Additionally, we now provide further results for math reasoning:
> > > >
> > > > | Method    | GSM8k    |  MATH500   |
> > > > |-------------|-------------|-------------|
> > > > |LLaMA3-8B-Instruct|
> > > > |Zero-shot CoT (T=0.2) | 70.0| 26.3 |
> > > > |Zero-shot CoT (T=0) | 72.6| 27.7|
> > > > | Four-shot CoT (T=0.2)| 72.4| 22.2|
> > > > | Four-shot CoT (T=0)| 73.7| 23.2 |
> > > > | SWAP (w/o discriminator) | 78.1|35.1 |
> > > > | SWAP | 82.7| 40.2 |
> > > >
> > > >
> > > > **2. In-context learning (ICL) prompt format:**
> > > > The ICL prompt format can impact the performance on coding tasks.
> > > > We show with HumanEval, variations in the ICL format can lead to notable changes in performance.
> > > >
> > > > However, we would like to emphasize the following points:
> > > >
> > > > - **Absence of an official ICL prompt for HumanEval:** [3] does not provide an official ICL prompt for HumanEval. The official Llama3 report [2] only includes 0-shot result.
> > > > - **Unique challenges in coding tasks:** Coding tasks differ from other benchmarks because the model often generates unnecessary code snippets, which cause parsing issues during evaluation. We invested significant effort in carefully adjusting the prompt for HumanEval to achieve higher performance.
> > > > - **Impact of test set size:** The HumanEval test set is relatively small (164 samples) compared to other benchmarks, which amplifies the impact of individual problems on accuracy. For example, a single problem corresponds to a 0.6% accuracy change in HumanEval, whereas this value is 0.07% in GSM8K and 0.02% in MATH.
> > > > - **Official ICL prompt available for other benchmarks:** For other benchmarks such as GSM8K and MATH, we follow publicly available official ICL prompts. Therefore, there are no similar issues for these benchmarks.
> > > >
> > > > **3. Setting selection standard:**
> > > > For methods other than ICL, we either follow the settings provided in the corresponding paper or official implementation, or we make our best effort to achieve higher performance while ensuring a fair comparison.
> > > >
> > > > Again, achieving the ‘best’ performance is non-trivial due to variations in both base models and benchmarks, which extend beyond the scope of our research.

---

> > > > > ### Author Response · Authors · 2024-12-01
> > > > > **Continuation of Rebuttal by Authors to Reviewer F56z**
> > > > >
> > > > > ## Comparison between PRM and SWAP
> > > > >
> > > > > We would like to emphasize the key differences (advantages) of SWAP over PRM:
> > > > >
> > > > > **1. Planning vs. Verification:** PRM is a verification method, meaning it only takes effect after the trajectory is complete. In contrast, SWAP is a planning method, where the discriminator actively guides the selection of actions mid-process, improving sampling efficiency.
> > > > >
> > > > > **2. Enhanced Generator:**
> > > > > Our generator is significantly stronger, as we fine-tune it on RL trajectories. Unlike CoT, we model the reasoning process as a MDP with interleaved actions and states, resulting in a more fine-grained training and inference process.
> > > > >
> > > > > The evidence is that the average token usage for CoT is 175.6 on GSM8k, whereas SWAP (without the discriminator, i.e., no planning) uses 306.9 tokens.
> > > > > As noted by OpenAI, there exists an inference-time scaling law: models achieve higher accuracy with more inference tokens, which explains SWAP's superior performance over CoT.
> > > > >
> > > > > Additionally, we incorporate diversity-based modeling to enhance sampling diversity and further improve efficiency.
> > > > >
> > > > > In contrast, the generator of PRM uses conventional CoT with IID sampling.
> > > > >
> > > > >
> > > > > **3. Structural Information:**
> > > > > A key innovation of our framework is treating multi-step reasoning as entailment graph construction.
> > > > > These structures explicitly represent statement dependencies, providing additional guidance for the model.
> > > > > They also enable structural verification (e.g., rule-based validity checks) to select better options.
> > > > > At a high level, this approach enforces rigorous reasoning while maintaining expressiveness.
> > > > >
> > > > > **4. Improved Discriminator:**
> > > > > Our framework enhances discrimination through contrastive ranking and meta-knowledge.
> > > > > Training an effective PRM is challenging, as it assigns independent numerical values within $[0, 1]$ to each state.
> > > > > In contrast, our discriminator evaluates relative quality, making it more robust to noise, simplifying optimization, and improving generalization.
> > > > > Furthermore, we enhance discrimination accuracy using meta-knowledge extracted from training samples, which identifies common pitfalls for specific problem classes.
> > > > >
> > > > > Given these significant differences (advantages), it is unsurprising that SWAP achieves substantial improvements over PRM. In fact, these results align with our expectations.
> > > > >
> > > > > We have restructured the implementation code for PRM and SWAP. We provide them for further verification.
> > > > >
> > > > > Anonymous link: https://drive.google.com/drive/folders/1ZvevZFc-LKYfCESvSdo3BUbvc94FoCPS?usp=sharing
> > > > >
> > > > > ## Reference:
> > > > >
> > > > > [1] https://github.com/meta-llama/llama3/blob/main/eval_details.md
> > > > >
> > > > > [2] https://github.com/meta-llama/llama3/blob/main/MODEL_CARD.md
> > > > >
> > > > > [3] https://github.com/openai/human-eval/tree/master

---

### Official Review · Reviewer_p8KY · 2024-11-04

**Soundness:** 3
**Presentation:** 3
**Contribution:** 2
**Rating:** 6
**Confidence:** 4

**Summary:**

This paper proposes SWAP, a framework to prompt GPT-4 generate solution chain with reflection to solve problems, and fine-tune other models on it to improve performance.

**Strengths:**

The writing is easy to follow. They show the prompts used and examples for different tasks in detail. They do ablation studies to show the effectiveness.

**Weaknesses:**

1. The baseline methods only include inference-time techniques without additional fine-tuning. How do these results compare to methods that incorporate fine-tuning? Specifically, I'm referring to approaches where GPT-4 generates or rewrites content based on the training set, and a base model is subsequently fine-tuned on those outputs.

2. What is the actual novelty of this work? Numerous studies already prompt GPT-4 with various roles for tasks like analysis, reflection, and planning.

3. How is the semantic similarity probability calculated? Also, the current normalization method seems weird. Is there a reason for mapping negative values to zero? This design choice seems suboptimal to me.

**Questions:**

What is the underlying connection to reinforcement learning in this approach? The process of constructing supervision data seems quite similar to the steps taken for creating training data for reward models in Math-Shepherd. Additionally, how does the discriminator here relate to the concept of a reward model?

Could you clarify what "world model" specifically refers to in this context? How is it defined, and why is it considered a world model?

---

> ### Author Response · Authors · 2024-11-22
> **Rebuttal by Authors to Reviewer p8KY**
>
> Thank you for your thoughtful comments, all of which are very helpful for improving our work!
>
> ## 1. Baseline with fine-tuning
>
> To investigate the impact of fine-tuning, we consider the following methods for comparison:
> 1) **SFT on CoT** (shown in Table 1): for datasets that provide reasoning process such as GSM8k and MATH, we directly fine-tune on it; for datasets without reasoning process such as FOLIO, ReClor, we use GPT-4o to generate the reasoning process data and fine-tune on CoTs that lead to the correct final answer. For coding datasets, we fine-tune on the completed code.
> 2) **SWAP w/o discriminator** (shown in Table 1): Since we simulate a Markov decision process (a sequence of interleaved states and actions) with structural information, which are different from CoTs, we consider the fine-tuned version of our framework with only the generator. Since there is no discriminator, we do not perform any planning during inference.
>
> Our findings include that fine-tuning on trajectories with structural information brings benefits than CoTs, but more importantly, planning during inference further improves the performance. It is supported by the performance comparison between **SWAP** and **SWAP w/o discriminator** in Table 1.
>
>
> ## 2. Novelty
> The key innovation in our framework is viewing multi-step reasoning as the process of **entailment graph construction**, supported by a **structure-aware planning** approach tailored for this purpose.
>
> Using graph representation offers several advantages: (1) It explicitly captures the **non-linear structure** of the reasoning process, providing LMs with enhanced guidance; (2) It allows for greater **control** over the reasoning process. For instance, graph representation facilitates structural verification during training data collection and enhances discrimination accuracy during inference. (3) It also enables exciting future work, such as exploring **the impact of action order** in multi-step reasoning. By capturing the dependency between steps, it becomes possible to identify interchangeable steps for data augmentation. Furthermore, for **long-context reasoning** tasks (e.g., OpenAI's o1 model), a graph structure can provide improved understanding and better control over the process.
>
> Given the representation, we further notice that existing agentic frameworks mainly use **prompt engineering** which totally or partially rely on **self-evaluation**. These strategies bring limited benefits in complex reasoning tasks since self-evaluation without external feedback can be unreliable [1,2]. To overcome this challenge, we propose using a **generator-discriminator** structure with fine-tuning, and further identify the bottlenecks of **generation diversity** and **discrimination accuracy**. We address these bottlenecks with **architecture-level adaptation**. All these strategies distinguish our framework from related work and contribute to **substantial improvements** (Table 1).
>
> ## 3. Semantic similarity probability calculation
> We add the calulation process in the revised version (Eq. 1).
>
> As for normalization, there are mainly two cases when tokens have **negative values** in the adjusted probability $P_{\pi{\text{G}}}$:
> 1) The token’s original probability $P_{\pi{G}}^{ori}$ is high, but its semantic similarity probability $P^{{sem}}_{\pi{{G}}}$ is even higher. This typically occurs for tokens that **resemble previous generations**, making them less desirable for diversity.
> 2) The token’s original probability $P^{{ori}}_{\pi{{G}}}$ is not high, indicating it likely represents a step that **deviates from the intended progression of reasoning**.
>
> In both cases, setting negative values to zero allows us to discard these tokens and focus on maintaining a diverse and relevant generation.
> Other alternatives, such as Softmax, can distort the probability of irrelevant tokens by redistributing values across all tokens, even those that should ideally have very low or zero probability.
>
> Consider an example with an intermediate step in a math problem:
>
> For the first token, the original probability distribution $P_{\pi{G}}^{ori}$ is: {"Let": 0.6, "Simplify": 0.2, "Consider": 0.2, every other token: 0}. Given the previous generation "Let $x$ be 4.", the semantically similar probability distribution $P_{\pi{G}}^{sem}$ for the first token becomes: {"Let": 0.5, "Set": 0.2, "Consider": 0.3, every other token: 0}. Let $\gamma=1$, then $P_{\pi{G}}^{ori} - \gamma P_{\pi{G}}^{sem}$ becomes: {"Let": 0.1, "Simplify": 0.2, "Consider": -0.1, "Set": -0.2, every other token: 0}.
> Using the proposed method yields {"Let": 0.33, "Simplify": 0.66, every other token: 0}, focusing on high-relevance tokens.
> However, applying Softmax produces {"Let": 0.00011, "Simplify": 0.00012, "Consider": 0.00009, "Set": 0.00008, every other token: 0.0001}, which spreads probability mass thinly across all tokens, diluting the relevance of the original distribution and ultimately harming model performance.

---

> > ### Author Response · Authors · 2024-11-22
> > **Continuation of Rebuttal by Authors to Reviewer p8KY**
> >
> > ## 4. Connection to RL
> > First, the main difference between our discriminator and process reward model such as Math-Shepherd is that we use **ranking**. The rationale here is that training an effective PRM remains challenging, as from a mathematical perspective, it assigns a **numerical value** within $[0, 1]$ to each state **independently**. To contrast, our ranking strategy offers significant advantages: 1) it emphasizes relative quality, making it more robust to noise; 2) it simplifies optimization and enhances generalization.
> > Notably, our high-quality **automatic ranking annotation** method is non-trivial as it systemically incorporates three key factors: 1) **structural information**; 2) **correctness**; and 3) **semantical equivalence**.
> >
> > Here, the policy discriminator in our framework, which selects actions based on the predicted future state, is conceptually similar to the Q-function in RL, i.e., the expected cumulative reward.
> >
> > The world model is defined as the state transition distribution. In RL, feedback is typically obtained directly from the environment. However, in some scenarios, collecting real-world feedback can be expensive or infeasible at scale. Consequently, recent research has focused on simulating the environment using a world model. In our framework, the world model is crucial for determining state (graph) transitions resulting from specific actions, guiding the reasoning process.
> >
> > ## Reference:
> > [1] Huang, Jie, et al. "Large language models cannot self-correct reasoning yet." arXiv preprint arXiv:2310.01798 (2023).
> >
> > [2] Jiang, Dongwei, et al. "Self-[in] correct: Llms struggle with refining self-generated responses." arXiv preprint arXiv:2404.04298 (2024).

---

> > > ### Author Response · Authors · 2024-11-25
> > > **Look forward to discussions**
> > >
> > > Thank you for your valuable feedback on our submission!
> > > We have carefully reviewed your comments and made substantial efforts to refine the paper and add additional experiments.
> > > We look forward to engaging in further discussions and addressing any potential remaining concerns in the days ahead.
> > > Thanks in advance!

---

> > > > ### Comment · Reviewer_p8KY · 2024-11-27
> > > >
> > > > Thank you to the authors for the detailed rebuttal. Some of my concerns have been addressed, so I am raising my score to 6.

---

> > > > > ### Author Response · Authors · 2024-11-27
> > > > > **Thanks for reply**
> > > > >
> > > > > Thank you for your recognition and support!

---

### Author Response · Authors · 2024-11-26
**Acknowledgment of Reviewer Contributions**

We sincerely appreciate the valuable comments and insights provided by all the reviewers. We have carefully reviewed the feedback to ensure a thorough understanding of the concerns raised. In response, we have refined our presentation and conducted additional experiments to address these concerns comprehensively. We are pleased with the substantial improvements our paper has achieved as a result of this process.

We are particularly grateful to Reviewers F56z, zS5k, and 38qG for engaging with our rebuttal and providing additional discussion. We also look forward to further interactions with Reviewer p8KY and are eager to address any remaining concerns from all reviewers.

We recognize ICLR's high standards and have made every effort to refine our paper to meet the expectations of the community. We sincerely hope the reviewers could consider our revised submission and the progress made during this process.

Thank the reviewers again for their time, insights, and constructive feedback!

---

### Meta-Review · Area_Chair_93np · 2024-12-19

**Metareview:**

The paper proposes a multi-step reasoning framework with methodological limitations. The Structure-aware Planning with Accurate World Model (SWAP) closely resembles existing approaches without presenting novel contributions. Reviewers highlighted persistent issues with presentation, notation clarity, and experimental validation. Performance improvements were marginal and inconsistent, with concerns about evaluation methodology and significant performance gaps. The implementation details remained vague, and authors incompletely addressed fundamental questions about the method's efficiency and generalizability. It is hard to say that the proposed approach substantially advances multi-step reasoning for large language models.

**Additional Comments On Reviewer Discussion:**

Reviewers raised critical concerns about the paper's methodology, including unclear notation, lack of novelty, and insufficient differentiation from existing approaches. Key issues involved ambiguous calculations, performance gaps, and vague implementation details. The authors responded by refining notations, adding pseudo-code, and conducting additional experiments with different models. They emphasized the framework's graph-based representation as a unique contribution. However, reviewers remained skeptical about the method's efficiency and fundamental innovation.

---

### Decision · Program_Chairs · 2025-01-22

Reject